# A Novel Heterogeneous Ensemble Framework Based on Machine Learning Models for Shallow Landslide Susceptibility Mapping

Haozhe Tang, Changming Wang *, Silong An, Qingyu Wang and Chenglin Jiang

College of Construction Engineering, Jilin University, Changchun 130012, China; tanghz21@mails.jlu.edu.cn (H.T.)
* Correspondence: wangcm@jlu.edu.cn

**Abstract:** Landslides are devastating natural disasters that seriously threaten human life and property. Landslide susceptibility mapping (LSM) plays a key role in landslide hazard management. Machine learning (ML) models are widely used in LSM but suffer from limitations such as overfitting and unreliable accuracy. To improve the classification performance of a single machine learning (ML) model, this study selects logistic regression (LR), support vector machine (SVM), random forest (RF), and gradient boosting decision tree (GBDT), and proposes a novel heterogeneous ensemble framework based on Bayesian optimization (BO), namely, stratified weighted averaging (SWA), to test its applicability in a typical landslide area in Yanbian Prefecture, China. Firstly, a dataset consisting of 1531 historical landslides was collected from field investigations and historical records, and a spatial database containing 16 predisposing factors was established. The dataset was divided into a training set and a test set in a ratio of 7:3. The results showed that SWA effectively improved the Accuracy, AUC, and robustness of the model compared to a single ML model. The SWA achieved the best classification results (Accuracy = 91.39% and AUC = 0.967). To verify the generalization ability of SWA, we selected published landslide datasets from Yanshan country and Yongxin country in China for testing. SWA also performed well, with an AUC of 0.871 and 0.860, respectively. As indicated by shapely values (SVs), Normalized Difference Vegetation Index (NDVI) is the factor that has the greatest impact on landslide occurrence. The landslide susceptibility maps obtained from this study will provide an effective reference program for land use planning and disaster prevention and mitigation projects in Yanbian Prefecture, China.

**Keywords:** landslide susceptibility mapping; ensemble learning; Bayesian optimization; machine learning models; shapely values

## 1. Introduction

According to the Critical Incident Database (https://www.emdat.be, accessed on 12 March 2023), landslides around the world have caused 66,438 deaths and economic losses of approximately USD 10.8 billion [1] during 1900–2020. Accurate prediction of the potential location of landslides can greatly minimize the losses caused by landslides. Landslide susceptibility prediction can effectively address this major issue based on the landslide inventory and related predisposing factors [2,3]. Therefore, landslide susceptibility mapping (LSM) is an important part of landslide prevention and management, which is often considered the first stage of disaster management and can provide scientific guidance for further disaster management [4,5]. Currently, various models have been designed based on GIS and machine learning (ML) techniques: from the initial statistical application models such as the Frequency Ratio (FR), Analytic Hierarchy Process (AHP), and Information Value Method (IVM) to the current Support Vector Machine (SVM) using kernel approach, Random Forest (RF) based on [6] Bagging, and XGBoost [7] based on Boosting.

Statistical methods are usually constructed using linear analysis between historical landslides and predisposing factors and are widely used in LSM due to their simple

principles and efficient computations. Rehman et al. [8] used Muzaffarabad as the study area and calculated the subjective and objective weights of all conditioning factors and their categories using AHP and FR. In addition, Lin et al. [9] explored the application of the IVM method in geological hazard evaluation to provide a scientific basis for the development of regional geological hazard prevention and control strategies. However, landslides occur under the coupling of multiple factors, and the data distribution of multiple contributing factors is different, so it is obvious that the relationship is not linear in the strict sense. Therefore, there are limitations to exploring LSM.

However, the ML method [10] can effectively overcome the effects of different data distributions and is good at capturing nonlinear relationships, improving the computational efficiency and accuracy of results by leaps and bounds. Models often used for LSM analysis are logistic regression (LR), support vector machine (SVM), random forest (RF), decision tree (DT), etc. Research results have been published on the application of these ML models in LSM. Chen et al. [11] used Chongren County, China, as the study area and combined it with 333 landslide sites for analysis, finally finding that RF has considerable performance in terms of AUC and statistical metrics. However, Hong et al. [12] showed that the predictive performance of LR is better than that of RF. Therefore, there are geographical differences in the prediction of LSM by different models, and there are limitations to improving the final classification effect by considering only the optimization of a single model. The ensemble methods [13] not only integrate the predictions of multiple models but are also more robust and easier to scale than a single model. The use of ensemble learning for LSM has been explored by scholars such as Lv et al. [14], who selected three base learners and four ensemble methods. The ensemble model will give better performance than a single model.

Therefore, for the application of EL in LSM, this paper proposes an ensemble framework for landslide susceptibility prediction in Yanbian Prefecture. In this paper, four of the most representative ML models, namely, LR, SVM, RF, and GBDT, are selected as single classifiers and combined with Bayesian optimization (BO) to propose a stratified weighted averaging (SWA) framework. This study has the following three main contributions. (1) It introduces BO into the hyperparameter optimization of a single model to improve the classification ability and robustness of a single model. (2) The four different ML models have significant differences in algorithm structure and feature processing, and SWA can reflect the performance improvement for a single model. At the same time, this paper comprehensively evaluates the performance of all models from multiple metrics, which can overall reflect the feasibility of all selected models. (3) Due to the uninterpretable characteristics of the ML training process, we cannot obtain the degree of influence of the selected conditional factors on the occurrence of landslides. Therefore, the SHAP method is selected to derive the factors that contribute most to the occurrence of landslides in the study area. Unlike previous related studies [15], this study obtains a set of model optimal parameter combinations after BO of a single model, and model derivation is performed on top of this to search for the best combination of weight values. On the one hand, the number of discriminative models is increased to improve the accuracy of prediction, and on the other hand, the computational process is simple and expandable.

## 2. Study Area and Dataset

### 2.1. Description of the Area

Yanbian Prefecture (Figure 1) is located in the Changbai Mountains region of Jilin Province, China, extending between longitudes of 127°27′E and 131°18′E and latitudes of 41°59′N and 44°30′N. The total area of Yanbian Prefecture is about 42,700 square kilometers, with the mountainous area accounting for 54.8% of the total area of the Prefecture, the plateau for 6.4%, the valley for 13.2%, the valley plain for 12.3%, and the hills for 13.3%. The overall topography of Yanbian Prefecture has a large elevation difference, with the west being higher and the east lower. The landforms are mountainous, hilly, and basin-like in 3 gradients from the southwest, northwest, and northeast to the southeast. The topography

of Yanbian Prefecture combines both terrain height and slope factors for the development of geological hazards.

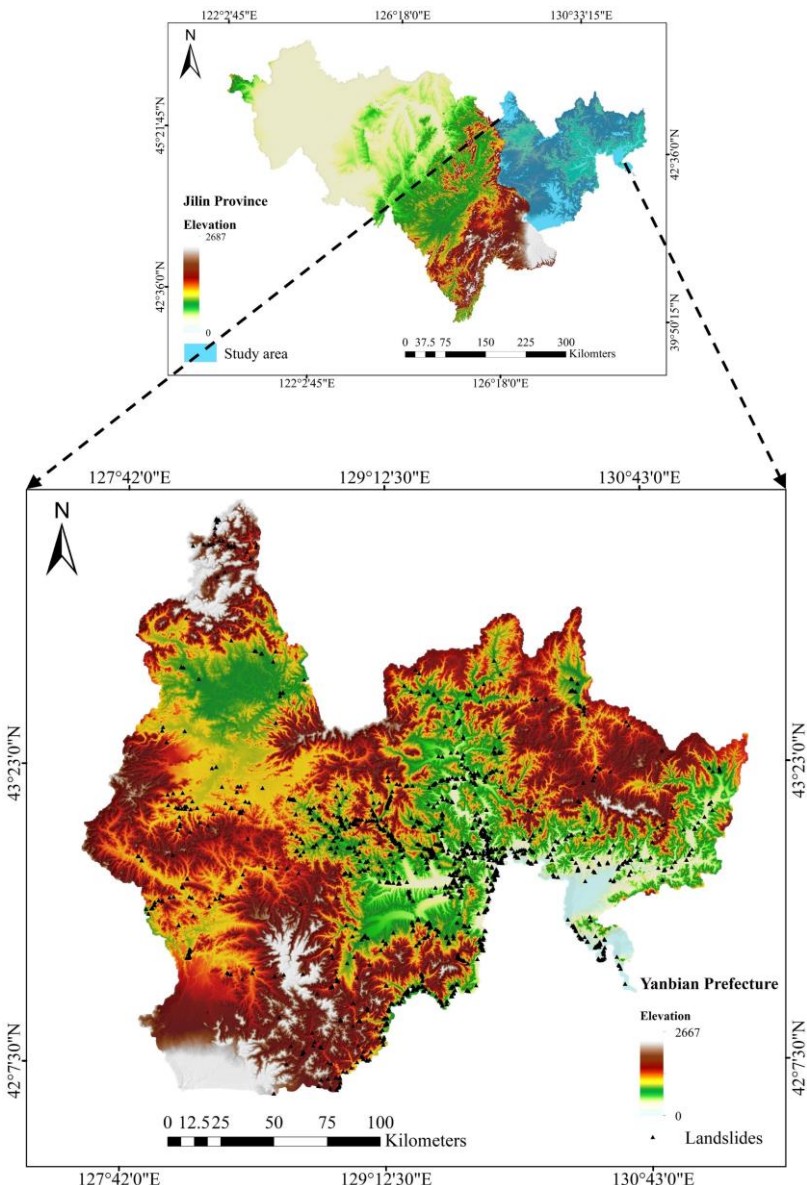

**Figure 1.** The map of study area.

The climate of Yanbian Prefecture is characterized by distinct monsoons, dry and windy in spring, warm and rainy in summer, cool and less rainy in autumn, and long cold periods in winter [16]. Precipitation spreads over a more uneven region, decreasing from the southeast to northwest. The leeward slopes of mountainous areas are smaller than the windward slopes, and areas close to the ocean with sufficient sources of water vapor are larger than those distant from the ocean with insufficient water vapor. The yearly average rainfall is mostly 550–650 mm, with rainfall concentrated from June to August, accounting for about 60% of the annual rainfall.

### 2.2. Landslide Inventory

Landslide inventories are the basis for landslide susceptibility studies. Landslide inventory mapping can enhance our understanding of the relationship between historical landslide spatial distributions and LSM [17,18]. The landslide inventory of Yanbian Prefecture is obtained by the Department of Natural Resources of Jilin Province through

field research and historical disaster records during 2010–2020. The landslides in the study area are mainly shallow landslides with a thickness of 0.5–10 m in the form of rotation or translation, most of which are related to the upper soil layer [19]. Landslides are mainly caused by rainfall.

Since landslides occur in a few areas and most areas are free of geological hazards, non-landslide points are selected at a ratio of the positive/negative samples of 1:1.15 to make the dataset more consistent with the actual geological environment. Meanwhile, according to the relevant studies [20], appropriately increasing the number of negative samples is beneficial to model training. Therefore, the dataset contains 1531 historical disaster points and 1761 non-landslide points that are randomly selected on top of the 400 m buffer zone of disaster points. The training and test sets are divided into a 7:3 ratio.

*2.3. Data Preparation*

Selecting the suitable predisposing factors is a key step in LSM [21]. However, in the absence of adequate corresponding assessment standards and technical specifications, the selection of landslide conditioning factors depends on subjective judgment. Reichenbach et al. [22] analyzed 565 relevant papers over the period of 1983–2016, from which they found that these factors can be broadly classified into 5 categories: geological, hydrological, land cover, geomorphological, and other. On the basis of a field survey, a literature review, and available data from the study area, the following 16 condition factors are selected. All data preparation is implemented in ArcGIS 10.8 software. All data sources and types can be seen in Table 1. Elevation, slope, aspect, profile curvature, Topographical Roughness Index (TRI), Topographical Wetness Index (TWI), and Stream Power Index (SPI) extracted from the Digital Surface Model (DSM) are provided by the ALOS Global Surface Model. Lineaments [23] include faults, ridgelines, fissures, or boundaries between strata on the Earth's surface, with different lengths and orientations. To extract the lineaments in the study area, the DSM is used to obtain 0°, 45°, 90°, and 135° mountain-shadow maps.

**Table 1.** Landslide predisposing factors and their type, source, and resolution (scale).

| Predisposing Factors | Data Type | Source | Scale/Resolution |
|---|---|---|---|
| Elevation | Raster | Derived from ALOS Global Surface Model 'ALOS World 3D-30 m' (http://www.eorc.jaxa.jp/ALOS/en/aw3d30/data/index.htm, accessed on 1 November 2022) | 30 m × 30 m |
| Slope | | | |
| Aspect | | | |
| Profile curvature | | | |
| TWI | | | |
| SPI | | | |
| TRI | | | |
| The density of lineaments | | | |
| Landform | Raster | Derived from A New Map of Global Ecological Land Units | 250 m × 250 m |
| Lithology | | Derived from A New Global Lithological Map Database | |
| NDVI | Raster | Derived from Resource and Environment Science and Data Center (https://www.resdc.cn/, accessed on 8 October 2022) | 30 m × 30 m |
| Yearly average rainfall | | Derived from Fine Resolution Mapping of Mountain Environment (http://digitalmountain.imde.ac.cn/home, accessed on 18 September 2022) | |
| Land cover | | Derived from ESRI (https://livingatlas.arcgis.com/landcover/, accessed on 15 October 2022) | 10 m × 10 m |

**Table 1.** *Cont.*

| Predisposing Factors | Data Type | Source | Scale/Resolution |
|---|---|---|---|
| Distance to faults | | Derived from Seismic Fault Active Survey Data Center (https://www.activefault-datacenter.cn/, accessed on 20 September 2022) | 1:500,000 |
| Distance to roads | Polygon | Derived from National Platform for Common Geospatial Information Services (https://www.tianditu.gov.cn/, accessed on 22 September 2022) | 1:500,000 |
| Distance to the water system | | | |

The raster data for lithology and landform are obtained from A New Map of Global Ecological Land Units [24] and The New Global Lithological Map Database [25]. Fault data are provided by the Seismic Fault Active Survey Data Center (SFACDC) with a scale of 1:500,000. Road and water system data are obtained from the National Platform for Common Geospatial Information Services (NPCGIS). Normalized Difference Vegetation Index (NDVI) is calculated for each annual year of Landsat 5/8 remote sensing images based on Google Earth Engine (GEE), and then the maximum NDVI is obtained for the location of each image pixel. Yearly average rainfall (YAR) is provided from Fine Resolution Mapping of Mountain Environment during 1991–2020. Landform, Lithology, and Land cover are categorical variables, so the nearest neighbor resampling method is used to unify the resolution to 30 m × 30 m.

Elevation [26] controls the direction of watercourses and the density of the drainage network and has a significant effect on soil moisture and slope (Figure 2a). The slope controls the shear forces acting on the slope and is one of the key causes of landslides (Figure 2b). Theoretically, the risk of landslides increases in areas with higher slopes [27,28] and decreases to zero in areas with slopes below 5°. Aspect is related [29] to the occurrence of landslides because slopes with different aspects are affected differently by precipitation and solar radiation (Figure 2c). Pro Cur (Figure 2d) and TRI (Figure 2g) [30,31] can effectively reflect the topographic complexity and changes in surface relief and erosion patterns. The presence of lineaments [32] in an area can lead to more water infiltration into joints and cracks, thus increasing the likelihood of landslides. For quantitative analysis, lineament density (LD) is used as the predisposing factor (Figure 2h). The development of cracks and shear cracks along the fracture zone [33], the formation of fault damage zones [34], and water infiltration can damage the structure of nearby geotechnical bodies, so the distance to faults (Figure 2k) should be taken into account. The relief of the topography also plays an important role in the occurrence of landslides. Landslides mainly occur [35] in the loose accumulation layer, but landslides at bedrock are relatively rare. Lithology (Figure 2j) reflects the [36] physical and mineralogical properties of geotechnical bodies in the study area.

There are many water systems in the study area, and the rivers near the water systems have a strong erosion effect on the geotechnical body, which can easily lead to landslides. And the closer the water system is, the more serious the erosion is, so DTW (Figure 2m) and SPI (Figure 2f) [37,38] are included in the analysis. TWI (Figure 2e) is a physical indicator of the effect of regional topography on runoff direction and accumulation. This index [39] helps to identify rainfall runoff patterns, potential areas of increased soil water content, and areas of ponding and quantifies the control of topography on underlying hydrological processes. Rainfall is a common and important trigger in landslide hazards. Strong rainfall [40] not only softens the sliding zone but also increases the self-weight of the slope body. And the rainwater (Figure 2p) infiltrates and transports along the joint fissures and remains for a long time, which makes the rock body soft and changes the internal stress state of the slope, thus causing landslides to occur.

It is noteworthy that previous relevant studies have rarely considered the effects of human activities, but such factors cannot be ignored when determining landslide susceptibility. Human construction activities [41] such as slope cutting, road building (Figure 2l),

and bridge building will change the internal structure of the geotechnical body, thus affecting the stability of the slope. LC (Figure 2o) reflects the situation of land use [42]. Vegetation cover areas, bare ground, and water bodies have an important influence on the stability of slopes. NDVI (Figure 2n) reflects the vegetation cover of the study area [43]. The presence of slope vegetation enhances the resistance of hillsides to landslides. After establishing a good spatial database, Z-Score [44] normalization is used to facilitate the training of machine learning (ML) modeling.

$$z = \frac{x - \mu}{\sigma} \tag{1}$$

where $\mu$ represents the overall mean and $\sigma$ represents the overall standard deviation.

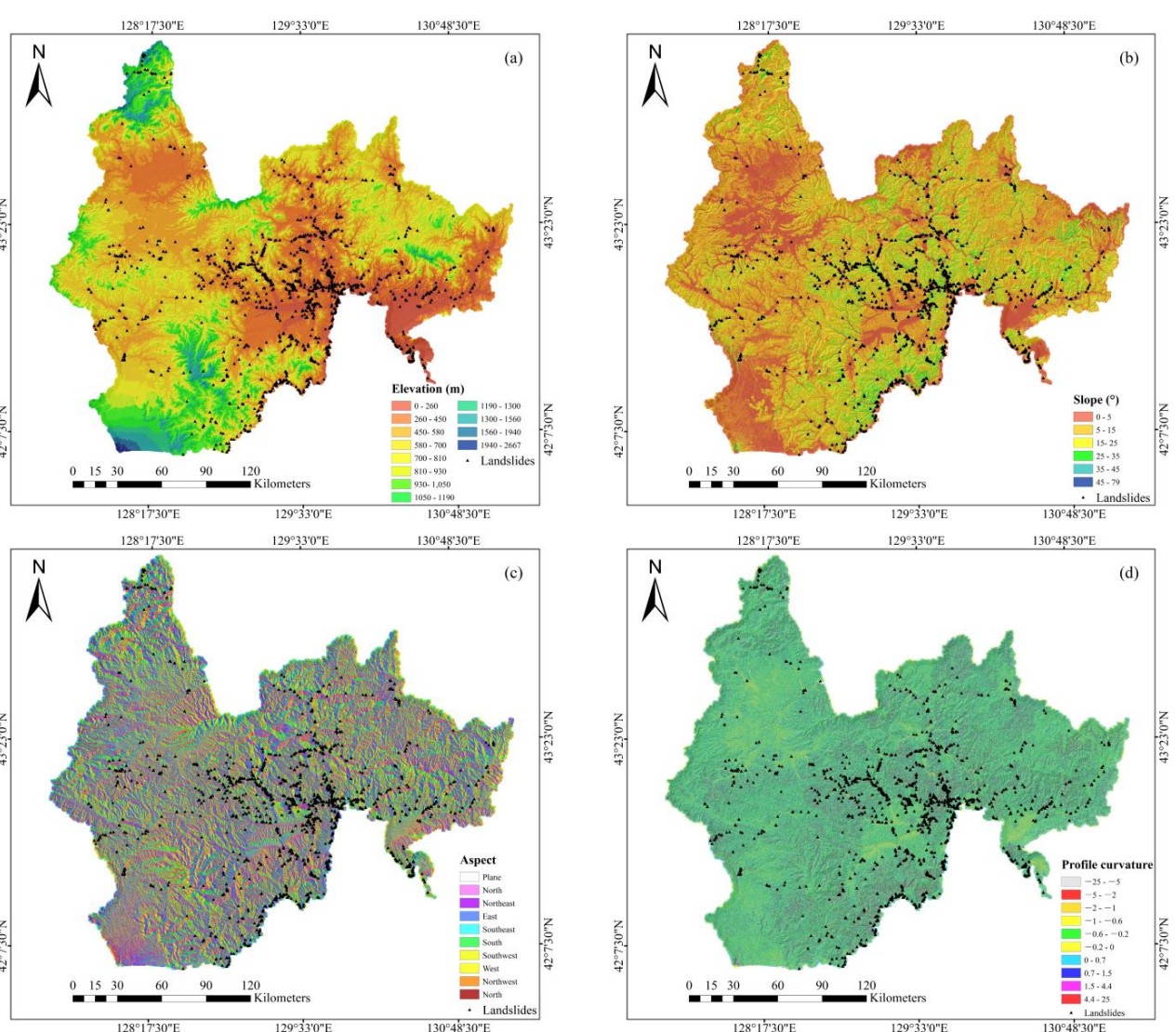

**Figure 2.** *Cont.*

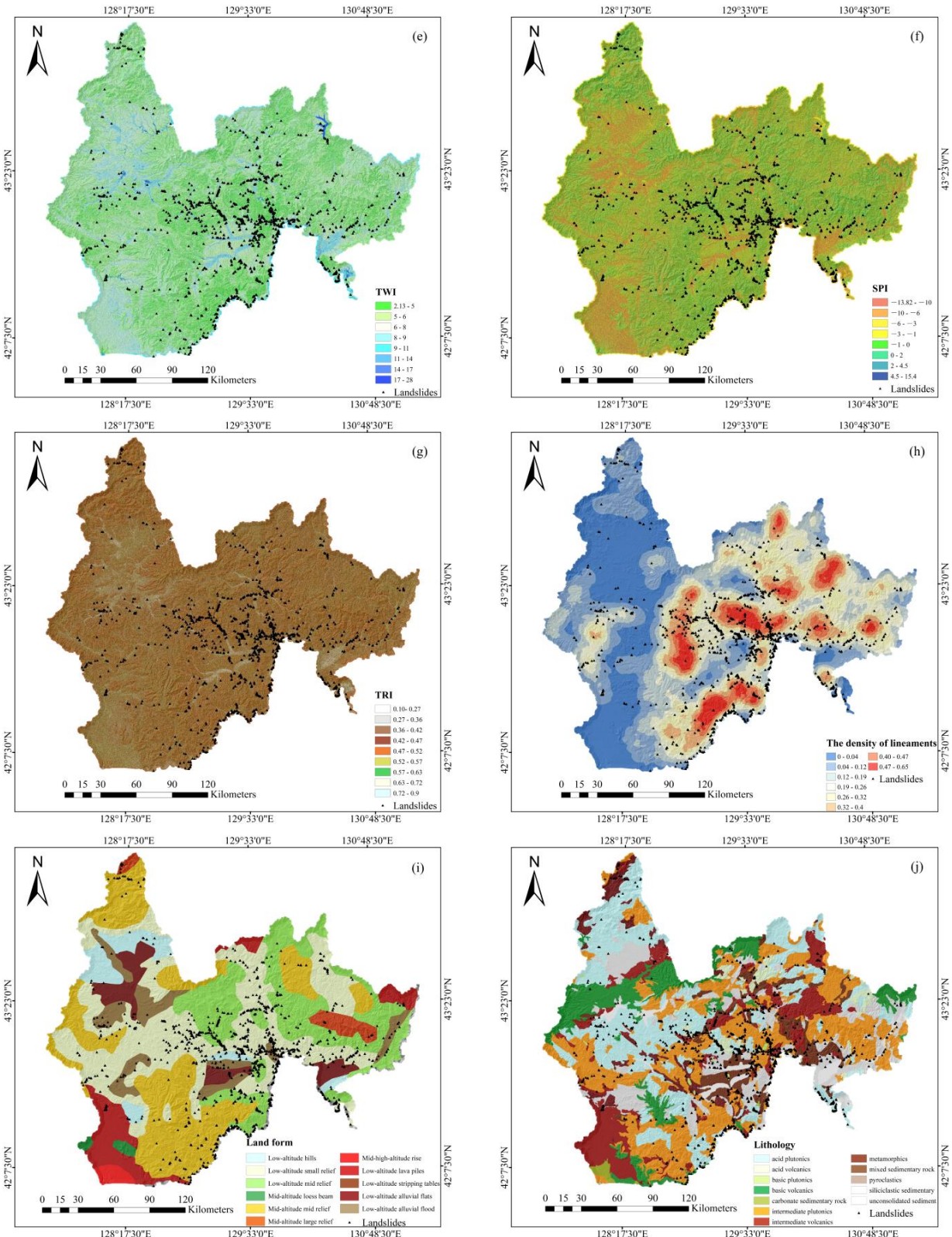

**Figure 2.** *Cont.*

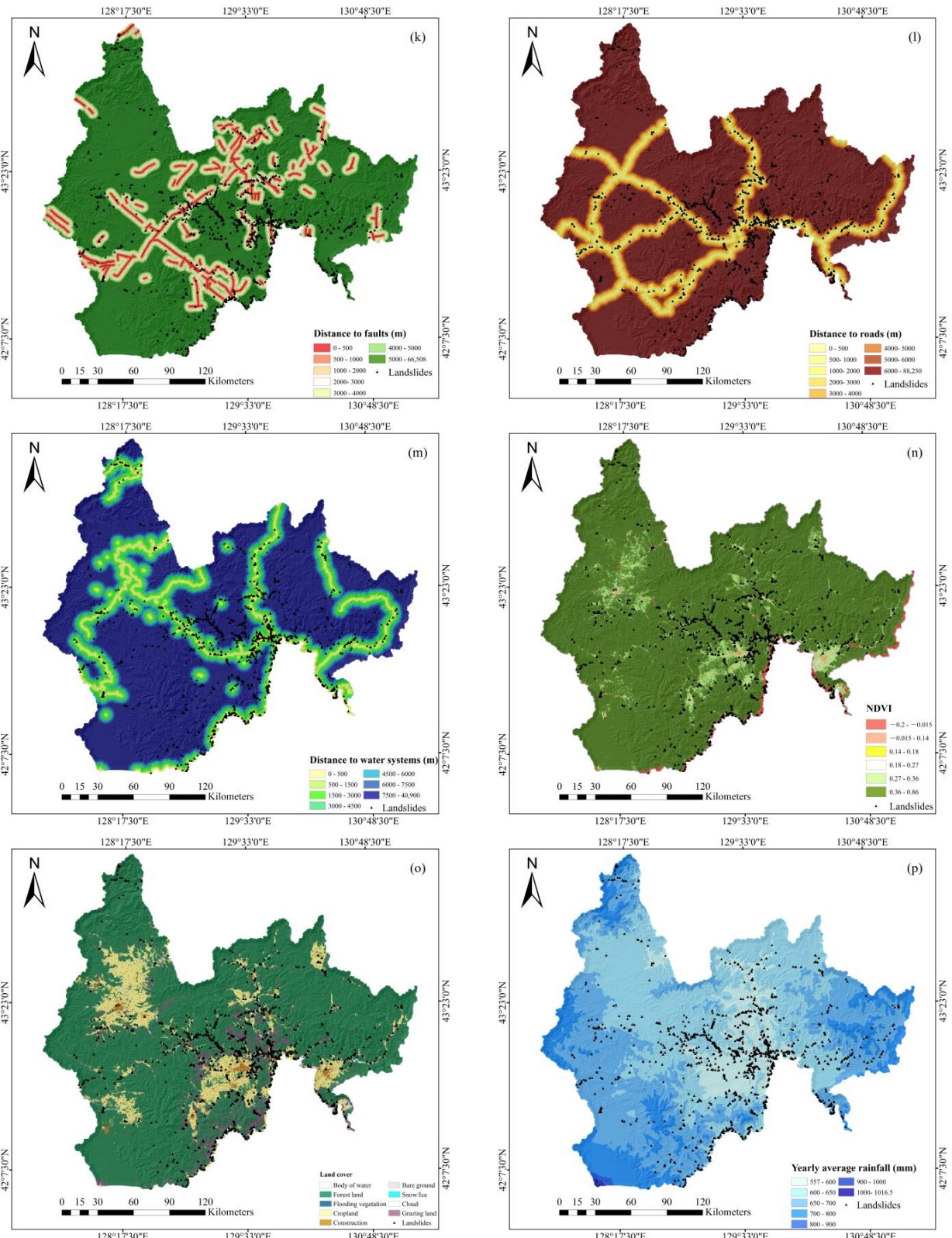

**Figure 2.** Maps showing landslide predisposing factors used in Yanbian Prefecture: (**a**) elevation, (**b**) slope, (**c**) aspect, (**d**) profile curvature (Pro_Cur), (**e**) TWI, (**f**) SPI, (**g**) TRI, (**h**) the density of lineaments (LD), (**i**) Landform, (**j**) Lithology, (**k**) distance to faults, (**l**) distance to roads, (**m**) distance to water systems, (**n**) NDVI, (**o**) Land cover (LC), and (**p**) Yearly average rainfall (YAR).

## 3. Methodology

Figure 3 shows the technical route, and this study includes the following steps. First, a dataset of the 16 predisposing factors mentioned in Section 2.3 and a dataset of hazard and non-hazard points are prepared. Secondly, 4 conventional ML models such as LR, SVM, RF, and GBDT are combined with BO for hyperparameter optimization, while a comparative study is conducted with the SWA model. Thirdly, to solve the black box problem of modeling, the SHAP method is introduced into the ML model interpretability analysis to evaluate the contribution value of each conditioning factor, which is introduced in Section 5.1.

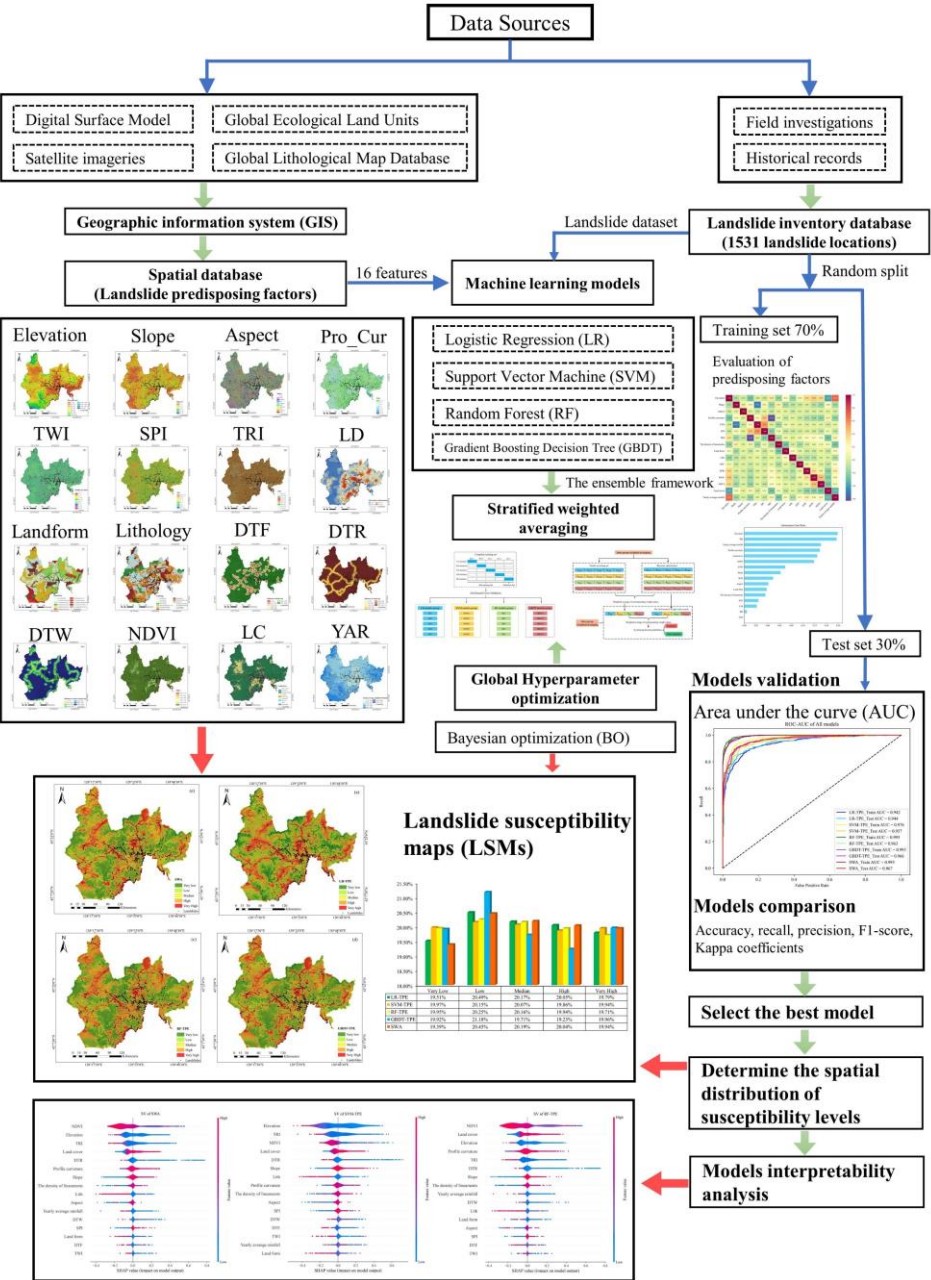

**Figure 3.** The flowchart of the developed methodology.

### 3.1. Evaluation of Predisposing Factors

3.1.1. Information Gain Ratio

To improve the training efficiency of the model, the first step should be to filter the selected elements. The information gain (IG) is introduced to measure how much

information the feature can bring to the classification task. The more information it brings, the more important the feature is. The expression of IG [45] is shown below:

$$I(X;Y) = H(X) - H(X|Y) = -\sum_x p(x)\log p(x) + \sum_y \sum_x p(y)p(x|y)\log p(x|y) \quad (2)$$

where H(X) represents the information entropy of X, and H(X | Y) represents the information entropy of the subset as a whole after the division based on different values of the Y array, also known as conditional entropy. The difference between the two is the information gain.

### 3.1.2. Multicollinearity Analysis

While considering the importance of the features, it is also necessary to consider the correlation between the features, and the metrics with too high a correlation should be eliminated to ensure the training efficiency of the model. To this end, the Pearson correlation coefficient is introduced [46]. This indicator is used to measure the covariance between variables X and Y, and its value range is $[-1, 1]$; $\rho = 0$ means no correlation between the two variables, and $\rho = 1$ or $\rho = -1$ means a linear relationship between the two variables. The expression is as follows:

$$\rho = \frac{Cov(X,Y)}{\sigma_X \sigma_Y} = \frac{E[(X - \mu_X)(Y - \mu_Y)]}{\sigma_X \sigma_Y} \quad (3)$$

where $Cov(X,Y)$ represents the covariance between the variables $X$ and $Y$; $\sigma_X$ and $\sigma_Y$ represent the standard deviation of the variables $X$ and $Y$. $\mu_X$ and $\mu_Y$ represent the means of $X$ and $Y$, respectively. $E(\cdot)$ represents the mathematical expectation. The strength of the correlation has the following divisions: very strong (0.9–1.0), high (0.7–0.9), moderate (0.4–0.7), low (0.2–0.4), and very weak (0.0–0.2).

### *3.2. Machine Learning Models*

### 3.2.1. Logistic Regression

Logistic regression (LR) is a model often used in supervised learning to solve dichotomous classification tasks [47]. It is often used in LSM, and the expression of logistic regression is shown below:

$$f(z) = \frac{1}{1 + e^{-z}} \quad (4)$$

where $z = w_1 \cdot x_1 + w_2 \cdot x_2 + ... + w_N ... x_N + b$ represents the weighted linear combination; $b$ represents the intercept term of this function; $w_N$ represents the coefficients representing each variable; $x_N$(N = 1, 2, ..., 16) represents the 16 selected predisposing factors; and $f(z)$ represents the probability value of landslide occurrence, taking values in the range of [0, 1].

### 3.2.2. Support Vector Machine

Support vector machine (SVM) is a supervised machine learning algorithm that uses kernel methods and has a very excellent classification effect. SVM is often used in LSM [48]. SVM classifies the model with the smallest possible classification error on the data by up-dimensioning the data and finding a hyperplane as a decision boundary in the distribution of data, using the distance from the sample to the hyperplane. And the loss function of the SVM is as follows:

$$min_{w,b,\zeta} \frac{||w||^2}{2} + C\sum_{i=1}^{n} \zeta_i, s.t. y_i(w \cdot \Phi(x_i) + b \geq 1 - \zeta_i), \zeta_i \geq 0 \quad (5)$$

where $||w||^2$ represents the weight vector $w = (w_1, w_2, ... w_i)$ of l2 parametrizations. The vector $w$ represents the weights. $\Phi(x_i)$ represents the selected kernel function, while $w \cdot \Phi(x_i) + b \geq 1 - \zeta_i$ represents the final chosen decision hyperplane. $\zeta_i$ is used as a slack variable to measure how much the instance allows for interval misclassification. And

$C$ is used as a penalty factor to control the penalty by weighing between minimization and $\zeta_i$, under joint action of the 3 parameters acting together, thus minimizing the overall loss function.

### 3.2.3. Random Forest

The basic unit of Random Forest (RF) is the decision tree [49]. Each decision tree is a classifier. N trees will have N classification results. RF integrates all category voting results, assigning the most voted category as the final output. A single decision tree takes cross-entropy or the Gini coefficient to calculate the label impurity for tree building. Information entropy is used to measure the degree of confusion in the data and can also be used to measure the label purity of a dataset. The higher the entropy value, the greater the amount of information contained. The Gini coefficient is used as a measure of purity by calculating 1 minus the sum of squares of $p(i|t)$. The smaller the Gini coefficient, the higher the label purity.

$$Entropy(t) = -\sum_{i=1}^{c} p(i|t)log_2 p(i|t) \tag{6}$$

$$Gini(t) = 1 - \sum_{i=1}^{c} p(i|t)^2 \tag{7}$$

### 3.2.4. Gradient Boosting Decision Tree

Gradient Boosting Decision Tree (GBDT) [50] is often used in LSM due to its excellent classification performance and robustness. Unlike RF, GBDT uses regression trees to solve classification problems. After multiple iterations, each classifier is trained based on the residuals of the previous classifier, and the accuracy of the final classifier is continuously improved by reducing the bias.

Specifically, based on the results of the previous weak evaluator $f(x)_{t-1}$, the loss function L(x,y) is computed. L(x,y) is used to adaptively influence the construction of the next weak estimator $f(x)_t$ and integrate the model output. The output result of evaluator is $H_t(x_i)$, the result of which is affected by the overall all weak estimators $f(x)_0 \sim f(x)_T$.

$$H_t(x_i) = H_{t-1}(x_i) + \eta f_t(x_i) \tag{8}$$

The Friedman Mean Square Error (F-MSE) [51] is the main impurity measure used in GBDT. F-MSE can be interpreted as the product of the summed mean of the sample sizes on the left and right leaf nodes and the squared mean squared error difference between the left and right leaf nodes. $w_L$ and $w_R$ represent the sample size on the left and right leaf nodes, respectively. $r_i$ is the residual on sample i. $\hat{y}_i$ is the predicted value of sample at the current sub-node. This approach allows impurities to fall more quickly, making the overall branching more efficient.

$$F - MSE = \frac{w_L w_R}{w_L + w_R} \cdot \left( \frac{\sum_L (r_i - \hat{y}_i)^2}{w_L} - \frac{\sum_R (r_i - \hat{y}_i)^2}{w_R} \right)^2 \tag{9}$$

### 3.3. Bayesian Optimization

Hyperparameter optimization [52] can have a significant impact on the performance of machine learning models, which mainly include various types of grid-based search methods (GS) [53], Bayesian optimization (BO), and genetic algorithms (GAs). GS methods use enumeration or random sampling to search for parameters through the defined parameter space, and the search process generates a significant amount of computation. Although genetic algorithms [54] can effectively improve model performance, they involve complex coding and decoding processes and depend on the merit of the initial population. BO [55]

uses different probabilistic surrogate models (PSMs) with acquisition functions (AFs) to find the optimal solution fast and accurately.

The basic flow of BO is as follows. Assume that f(x) is a smooth and uniform function. First, two real observations are selected and the observation curve f* is inferred based on the PSM and contains the confidence level of this curve. Based on the PSM output, the maximum value of AF is calculated, and the x pointed by the maximum value is the next actual observation point. The estimated f* is continuously corrected by multiple observations. After a set number of estimations, f* continues approximating f(x), and then the minimum value of the final f* is taken as the minimum value of f(x).

In this paper, the tree-structure parameter estimator (TPE) is used as PSM and AF is used as the expected increment. The TPE process [56] divides $p(x|y)$ into two parts, and the equation is shown below:

$$p(x|y) = \begin{cases} \updownarrow(x) & \text{if } y < y^* \\ g(x) & \text{if } y \geq y^* \end{cases} \tag{10}$$

TPE constructs different distributions for observation point $x$ on either side of the threshold $y^*$, where $l(x)$ is the probability density function formed using the observation variable $x^{(i)}$ such that $y^* > y^{(i)}$. $g(x)$ is the probability density function using the remaining observations. The threshold for delineation is the quantile $\gamma$, so we have $p(y^* < y^{(i)}) = \gamma$. Therefore, we can obtain $p(x)$ and the expression for the collection function.

$$p(x) = \gamma l(x) + (1 - \gamma)g(x) \tag{11}$$

$$\text{EI}_{y^*}(x) = \int_{-\infty}^{\infty} \max(y^* - y, 0)p(y \mid x)dy = \frac{\int_{-\infty}^{y^*}(y^* - y)p(y \mid x)dy}{\propto \left(\gamma + (1 - \gamma)\frac{g(x)}{l(x)}\right)} \tag{12}$$

*3.4. Stratified Weighted Averaging*

Weighted averaging (WA) [57] is a common ensemble method that is widely used in landslide susceptibility prediction. This study aims to proposes an improved ensemble method based on WA. The stratified weighted averaging (SWA) is implemented in 3 steps. The whole process searches the weight values as hyperparameters to obtain the final prediction probability. First, the training set is divided into 5 copies, each containing 1 validation set and 4 sub-training sets (Figure 4). On top of the optimal combination of parameters for BO, the hyperparameters are adjusted using a grid search for 5 rounds of training. For a single classifier, 5 derived models can be obtained.

$$X_{LR} = \frac{x_{LR1} \cdot w_{LR1} + x_{LR2} \cdot w_{LR2} + x_{LR3} \cdot w_{LR3} + x_{LR4} \cdot w_{LR4} + x_{LR5} \cdot w_{LR5}}{w_{LR1} + w_{LR2} + w_{LR3} + w_{LR4} + w_{LR5}} \tag{13}$$

$$X_{SVM} = \frac{x_{SVM1} \cdot w_{SVM1} + x_{SVM2} \cdot w_{SVM2} + x_{SVM3} \cdot w_{SVM3} + x_{SVM4} \cdot w_{SVM4} + x_{SVM5} \cdot w_{SVM5}}{w_{SVM1} + w_{SVM2} + w_{SVM3} + w_{SVM4} + w_{SVM5}} \tag{14}$$

$$X_{RF} = \frac{x_{RF1} \cdot w_{RF1} + x_{RF2} \cdot w_{RF2} + x_{RF3} \cdot w_{RF3} + x_{RF4} \cdot w_{RF4} + x_{RF5} \cdot w_{RF5}}{w_{RF1} + w_{RF2} + w_{RF3} + w_{RF4} + w_{RF5}} \tag{15}$$

$$X_{GBDT} = \frac{x_{GBDT1} \cdot w_{GBDT1} + x_{GBDT2} \cdot w_{GBDT2} + x_{GBDT3} \cdot w_{GBDT3} + x_{GBDT4} \cdot w_{GBDT4} + x_{GBDT5} \cdot w_{GBDT5}}{w_{GBDT1} + w_{GBDT2} + w_{GBDT3} + w_{GBDT4} + w_{GBDT5}} \tag{16}$$

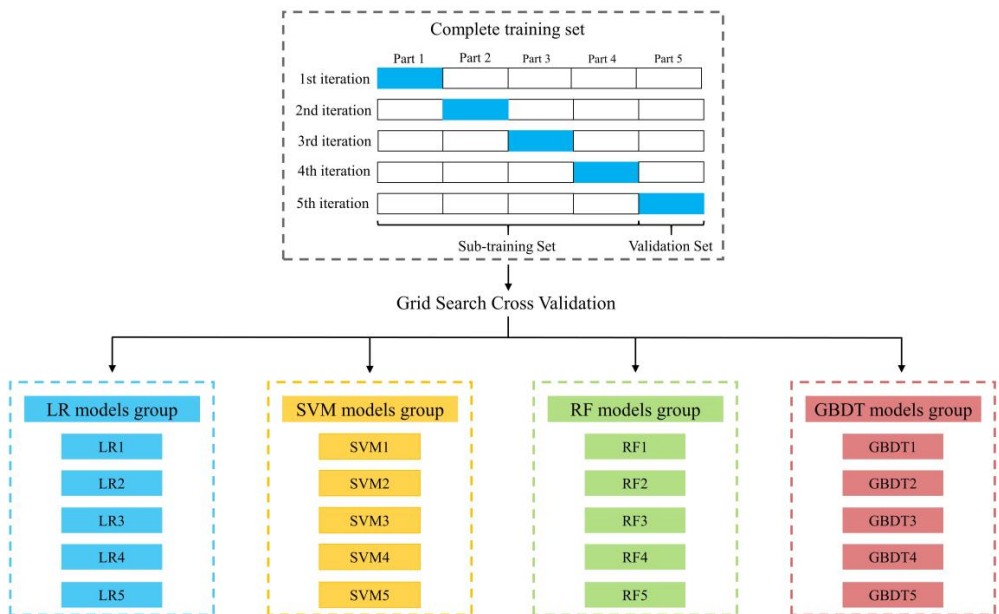

**Figure 4.** Model derivation.

Next, an intra-group fusion of predicted probabilities is performed (Figure 5). Taking LR as an example, the predicted probability ($X_{LR}$) of the LR model is obtained by predicting the 5 derived models ([LR1, LR2, LR3, LR4, LR5]) on the testing set, using BO search weight values. $x_{LR1}$ represents the predicted probability of LR1 on the testing set, and $w_{LR1}$ represents the weight value of $x_{LR1}$. And similarly, $X_{SVM}$, $X_{RF}$, and $X_{GBDT}$ represent the predicted probabilities of SVM, RF, and GBDT on the testing set, respectively. As a result, the predicted probabilities of intra-group fusion are obtained for all 4 base classifiers. Finally, the weight values of inter-group fusion are obtained by BO to obtain the final predicted probability ($X_P$), which yields the classification result. The judgment threshold is also searched as a hyperparameter and the final judgment threshold obtained is 0.5688.

$$X_P = \frac{X_{LR} \cdot w_{LR} + X_{SVM} \cdot w_{SVM} + X_{RF} \cdot w_{RF} + X_{GBDT} \cdot w_{GBDT}}{w_{LR} + w_{SVM} + w_{RF} + w_{GBDT}} \tag{17}$$

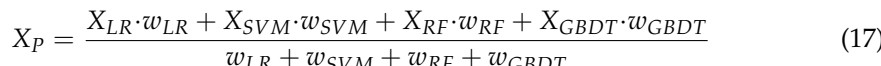

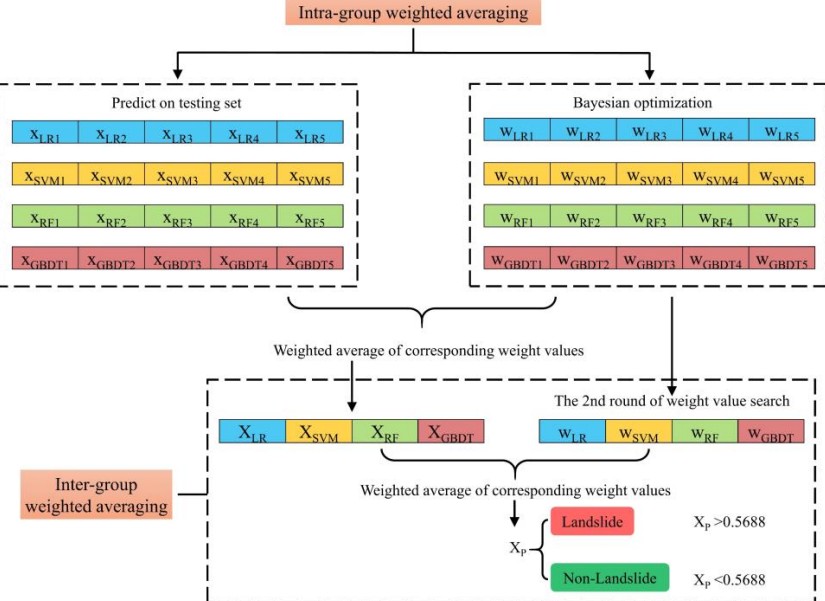

**Figure 5.** The framework of stratified weighted averaging.

### 3.5. Model Evaluation Metrics

LSM needs to be validated to have scientific significance, so it is necessary to select the appropriate indicators to evaluate the selected model. The ROC curve [58] is used as an important indicator of the overall classification effectiveness of the model. The horizontal coordinate of the ROC curve is the false positive rate, which represents the proportion of false samples among all the samples predicted to be of class 0. The vertical coordinate is the recall rate, which focuses on the proportion of all class 1 samples that are not accurately identified. The AUC is divided into 4 classes: poor (0.5–0.7), moderate (0.7–0.85), good (0.85–0.95), and excellent (0.95–1.00).

We obtain the confusion matrix [59] by comparing the predicted labels of the model in the testing set with the true labels based on the number of samples with false positive (FP), false negative (FN), true positive (TP), true negative (TN). TS is the sum of the number of TP, TN, FP, and FN samples. On this basis, Accuracy, Recall, and Precision are calculated. Meanwhile, to weigh the prediction results against the test set regarding the proportion of class 1 samples, we consider using the harmonic mean of Precision and Recall, i.e., F1-Score, as the model evaluation metric [60]. Finally, to measure the consistency between the predicted and true values, the Kappa coefficient is introduced. The more unbalanced the confusion matrix, the higher the $P_{exp}$ and the lower the Kappa value. The Kappa coefficients' [61] range is [−1, 1] and can be divided into 5 groups to represent different levels of consistency: slight (0.00–0.20), fair (0.21–0.40), moderate (0.41–0.60), substantial (0.61–0.80), and almost perfect (0.81–1.00).

$$Accuracy = \frac{TP + TN}{TP + TN + FP + FN} \tag{18}$$

$$Precision = \frac{TP}{TP + FP} \tag{19}$$

$$Recall = \frac{TP}{TP + FN} \tag{20}$$

$$F1 - Score = \frac{2}{\frac{1}{Recall} + \frac{1}{Precision}} = \frac{2 \cdot Recall \cdot Precision}{Recall + Precision} \tag{21}$$

$$Kappa = \frac{P_{obs} - P_{exp}}{1 - P_{exp}} \tag{22}$$

$$P_{obs} = \frac{TP + TN}{TS} \tag{23}$$

$$P_{exp} = \frac{(TP + FP)(TP + FN) + (FN + TN)(FP + TN)}{TS \times TS} \tag{24}$$

## 4. Result

### 4.1. Factor Assessment Results

Figure 6a shows the information gain value (IG) of each landslide conditioning factor. The top three IG values are elevation, TRI, and yearly average rainfall, which correspond to the complex topography of Yanbian Prefecture; the following factors are Land cover and NDVI, which reflect the influence of ecological environment on the occurrence of landslides, and places with high vegetation cover and good soil and water conservation will inhibit the occurrence of landslides; TWI is ranked last with IG values of 0, but TWI is an essential indicator of geological elements and topography analysis. So, it is not excluded.

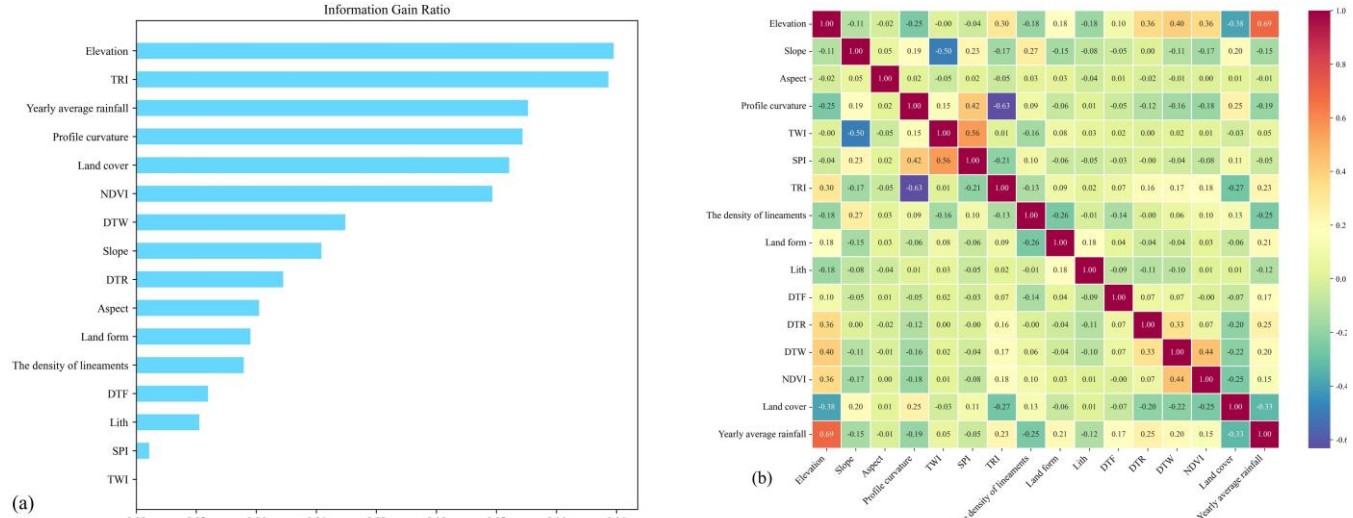

**Figure 6.** (**a**) Importance analysis; (**b**) correlation coefficient matrix.

As shown by the Pearson correlation coefficient graph (Figure 6b), the correlation coefficients of any two groups of 16 selected factors are less than 0.7, and there is no strong correlation. In summary, all 16 selected factors are used in LSM.

### 4.2. Landslide Susceptibility Maps

There are three main steps in making LSM [62]. Firstly, all the predisposing factors are added to the pixel cells of the whole study area, and then the probability of landslide occurrence is predicted separately under different models. Finally, the study area is classified into five classes using the quantile method, namely, very low, low, medium, high, and very high. The percentages of very low and very high prediction results (Figure 7) show a similar distribution pattern in the LSM of the five models.

The predicted results are consistent with the geographic characteristics of the study area. Very high areas are mostly areas with low elevation, while areas with high vegetation cover are mostly very low areas. According to the distribution results of LSM, the prediction results of RF-TPE are mostly distributed in low, median, and high classes, and the prediction results of GBDT-TPE are the opposite. SWA considers the prediction results of the four models simultaneously and assigns different weights to make the results of LSM more reliable.

The percentages of landslide susceptibility classes under the five models are shown in Figure 8. For the LSM generated by the LR-TPE model, the percentages of pixels occupied by very high and very low grades are 19.51% and 19.79%, respectively, while low, medium, and high occupy 20.49%, 20.17%, and 20.05%, respectively. For the LSM generated by SVM-TPE, the percentages of pixels occupied by very high and very low grades are 19.97% and 19.94%, respectively, while low, medium, and high occupy 20.15%, 20.07%, and 19.86%, respectively. For the LSM generated by RF-TPE, the percentage of pixels occupied by very high and very low grades are 19.95% and 19.71%, respectively, while low, medium, and high occupy 20.25%, 20.16%, and 19.94%, respectively. For the LSM generated by GBDT-TPE, the percentages of pixels occupied by very high and very low grades are 19.92% and 19.96%, respectively, while low, medium, and high occupy 21.18%, 19.71%, and 19.23%, respectively. For the LSM generated by SWA, the pixels occupied by very high and very low grades are 19.39% and 19.94%, respectively, while low, medium, and high occupy 20.45%, 20.19%, and 20.04%, respectively. The above results indicate that the high-susceptibility areas are concentrated in the lower-terrain areas, which are characterized by frequent human activities, and land cover type is mostly cropland. Meanwhile, according to the DTR (Figure 2l), landslides are densely distributed near the roads, but the areas with higher

vegetation cover are usually lower-susceptibility. Combining geomorphology, land cover, and human activities, the susceptibility distribution has geomorphic plausibility [63].

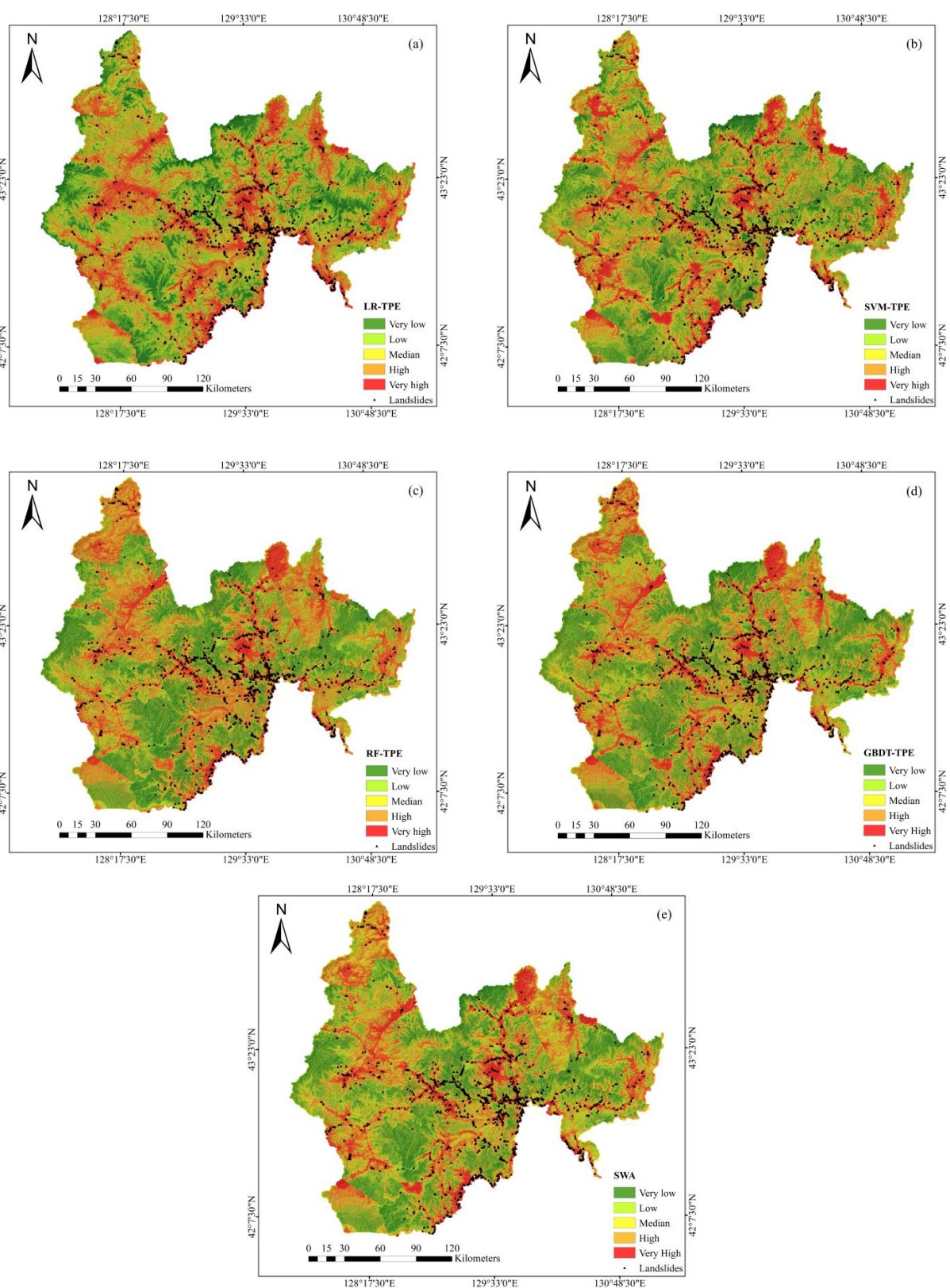

**Figure 7.** Landslide susceptibility mapping for 5 models (**a**–**e**).

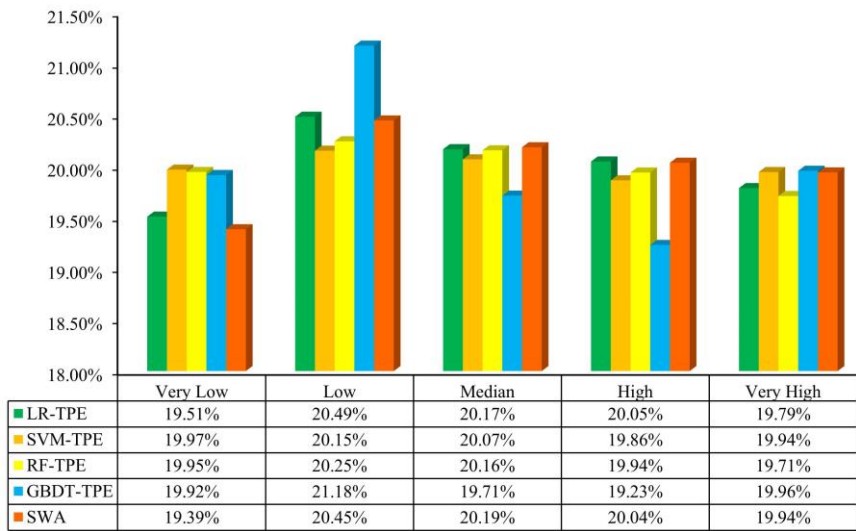

**Figure 8.** Percentage of susceptibility level.

*4.3. Model Validation and Comparison*

The evaluation of model performance plays a key role in LSM. In this study, four ML models are selected as the base classifier, and BO is used to obtain their optimal parameter combinations, on top of which the SWA method is used for model fusion. There are 1531 landslide points and 1761 non-landslide points in the study area, and the dataset is divided into a training set and a testing set according to the ratio of 7:3. In this paper, six frequently used model performance metrics are selected for comparative study. All models are built and optimized using the Python libraries: Scikit-Learn 1.0.1 and Hyperopt 0.2.5. All assessment metrics are obtained through five rounds of cross-validation [64].

Considering the classification effect of landslides and non-landslides (Table 2), GBDT-TPE (90.88%) outperforms the other models in terms of accuracy, followed by RF-TPE with SVM-TPE (90.37%) and LR-TPE (88.15%). In terms of the classification accuracy of landslide points in prediction results, SVM-TPE achieves the best Precision (90.20%), followed closely by GBDT-TPE (89.11%), then RF-TPE (88.45%) and LR-TPE models (85.40%). In terms of the accuracy of landslide point classification in the test set, GBDT-TPE obtains the highest Recall (91.09%), followed by RF-TPE (Recall = 90.63%), SVM-TPE (89.22%), and LR-TPE (88.69%). In terms of F1-Score, GBDT-TPE obtains the highest F1-Score (0.901), followed by SVM-TPE (Recall = 0.897), RF-TPE (0.895), and LR-TPE (0.870). The Kappa coefficient is used to measure the prediction accuracy and the best performance is obtained for GBDT-TPE and RF-TPE (0.817), followed by SVM-TPE (0.807) and finally LR-TPE (0.761).

**Table 2.** List of model metrics.

| Measures/Methods | LR-TPE | SVM-TPE | RF-TPE | GBDT-TPE | SWA |
|:---:|:---:|:---:|:---:|:---:|:---:|
| TP | 392 | 414 | 411 | 409 | 408 |
| FN | 50 | 50 | 42 | 40 | 34 |
| FP | 67 | 45 | 48 | 50 | 51 |
| TN | 478 | 478 | 486 | 488 | 494 |
| Accuracy | 88.15% | 90.37% | 90.37% | 90.88% | 91.39% |
| Recall | 88.69% | 89.22% | 90.63% | 91.09% | 92.31% |
| Precision | 85.40% | 90.20% | 88.45% | 89.11% | 88.91% |
| F1-Score | 0.870 | 0.897 | 0.895 | 0.901 | 0.906 |
| Kappa | 0.761 | 0.807 | 0.817 | 0.817 | 0.827 |

Finally, we compare the single model with SWA (Figure 9). Except for Precision, the rest of the metrics achieve the best performance with AUC = 0.967 and Accuracy = 91.39%, while the overfitting is greatly alleviated. This makes the LSM more realistic, while the robustness of the model is improved.

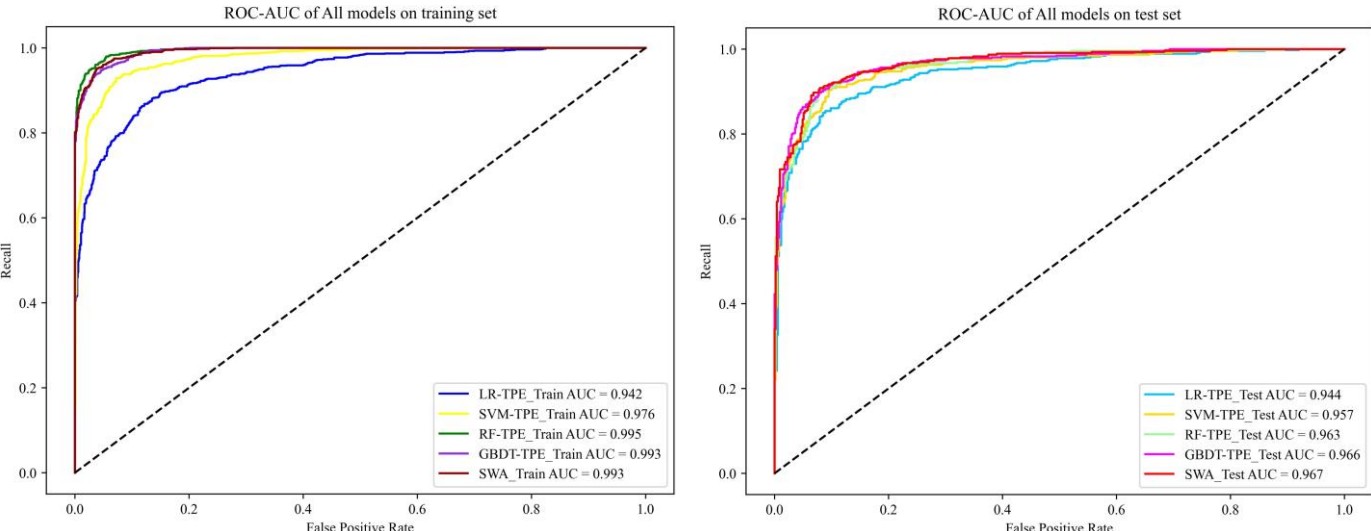

**Figure 9.** The ROC curves of all models.

## 5. Discussion

### 5.1. Analysis of Model Interpretability

The SHAP method [65] is often used to solve ML model interpretability problems. In interpreting the ML models, the SHAP values (SVs) can be understood as the importance of the contribution of the selected factors to the predicted values of the model. In the SHAP summary plots, the vertical coordinates represent the ranking of feature importance, from top to bottom, from most important to least important. The horizontal axis indicates the SV size of the selected factors [66], and the color of the selected factors represents the size of the value taken by the factor. When SV > 0, the larger the SV value, the more it promotes the occurrence of landslides, and conversely, when SV < 0, the smaller the SV, the more it inhibits the occurrence of landslides.

$$shape(X_i) = \sum_{s \subseteq N \setminus \{i\}} \frac{k!(p-k-1)!}{p!} (f(S \cup \{i\}) - f(S)) \tag{25}$$

The Shapley value for feature $X_i$ in a model is shown above, where p is the total number of features, $N \setminus \{i\}$ is a set of all possible combinations of features excluding $X_i$, $S$ is a feature set in $N \setminus \{i\}$, $f(S)$ is the model prediction with features in $S$, and $f(S \cup \{i\})$ is the model prediction with features in $S$ plus feature $X_i$. In terms of topographic factors, elevation has the highest SV value (Figure 10). It is worth mentioning that this is not to suggest that elevation is a trigger condition for landslides to occur, but in the area of lower elevation, the terrain is flat, human engineering activities are frequent, and cutting slopes to build roads will destroy the original geotechnical structure. At the same time, there is little vegetation cover, soil erosion is serious, and erosion of geotechnical bodies is aggravated by heavy rainfall, which very easily causes damage. The lower-elevation area is coupled with many landslide susceptibility factors. In terms of environmental factors, NDVI has the highest SV value. Lower vegetation cover or near-water bodies [67] are mostly high-landslide-susceptibility areas, while higher NDVIs are usually low-landslide-susceptibility areas. In terms of ecological factors, Land cover (LC) has the highest SV value. LC reflects the interaction between human activities and the natural environment [68], and areas with dense built-up areas or areas with dense cropland are usually high-susceptibility areas, while shrub forest areas are usually low-susceptibility areas. In summary, NDVI is the most

influential factor for landslide occurrence. It is recommended to build support projects in landslide-affected areas to prevent residents from being injured by falling rocks, with emphasis on ecological prevention and control to maintain soil and water stability [69]. After incorporating the spatial database of landslides, a geohazard project with monitoring and early warning was established to determine the status of landslides and set the warning level for the reference of the local government [70].

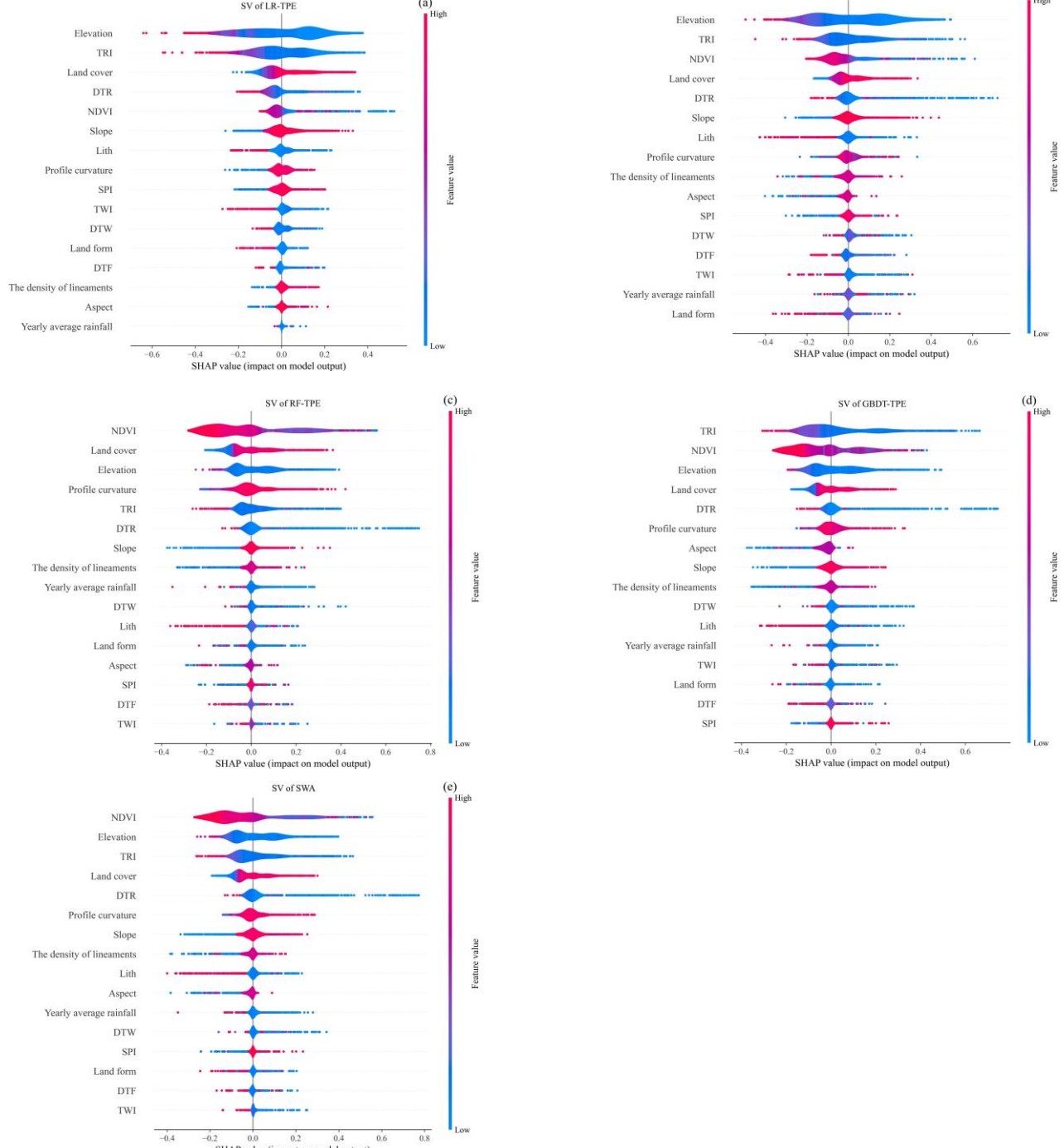

**Figure 10.** SV for 5 models: (**a**) LR-TPE, (**b**) SVM-TPE, (**c**) RF-TPE, (**d**) GBDT-TPE, and (**e**) SWA.

However, factors such as DTF and yearly average rainfall, which are crucial for landslide susceptibility studies, ranked low instead. Such factors require the necessary attention. Landslide susceptibility analysis requires comprehensive consideration.

*5.2. Model Optimization*

The model optimization in this study is divided into two parts: hyperparametric optimization of a single model based on BO, and the SWA that integrates the prediction results of multiple models.

There are five main parameters to be adjusted in the LR model (Table 3). 'penalty' adds regular terms to limit model complexity, control empirical and structural risks, and ultimately alleviate the overfitting. 'C' represents the weight of the empirical risk and structural risk of the model in the loss function. 'tol' sets the iteration stopping condition. 'max_iter' represents the maximum number of iterations when solving the parameters. 'solver' is the solution method of the loss function, which can satisfy both regularized solution cases. The final combination of the selected parameters is ['C' = 0.75, 'max_iter' = 400, 'penalty' = 'l1', solver = 'liblinear', 'tol' = 0.0977].

**Table 3.** The parameter space of LR.

| Parameters | 'penalty' | 'C' | 'tol' | 'max_iter' | 'solver' |
|---|---|---|---|---|---|
| Selected range | ['l1', 'l2'] | [0.5, 3] | [0.0001, 0.1] | [100, 10,000] | ['liblinear', 'saga'] |

There are six main parameters to be adjusted in the SVM model (Table 4). 'kernel' maps low-dimensional data to a high-dimensional space, thus enabling the classification task. 'degree' determines the maximum number of polynomials. 'gamma' affects the kernel function. In addition, 'class_weight' determines the adjustment of the model itself for the sample weights. C, which is the penalty term coefficient of the relaxation coefficient, determines the size of the decision boundary. 'coef0' is the independent term of the kernel function. The final selected combination of parameters is ['C' = 1.375, 'class_weight' = 'balanced', 'coef0' = 2.532, 'degree' = 2, 'gamma' = 'auto', 'kernel' = 'poly'].

$$K(x,y) = (\gamma(x \cdot y) + r)^d \tag{26}$$

$$K(x,y) = e^{-\gamma\|x-y\|^2}, \gamma > 0 \tag{27}$$

**Table 4.** The parameter space of SVM.

| Parameters | 'kernal' | 'degree' | 'gamma' | 'class_weight' | 'C' | 'coef0' |
|---|---|---|---|---|---|---|
| Selected range | ['poly', 'rbf'] | [0, 5] | ['auto', 'scale', [0.0001, 3]] | [None, 'balanced'] | [0.001, 2] | [2, 7] |

There are five tuning parameters for the RF model (Table 5). 'n_estimators' is the parameter with the greatest impact on the model, representing the number of evaluators involved in modeling. 'criterion' determines the branching criterion. 'max_depth' represents the maximum depth allowed for weak estimators. 'max_features' determines the number of randomly selected features. 'min_samples_split' limits the number of leaves and branches. The final selected parameter combination is ['max_depth' = 24, 'max_features' = 3, 'min_impurity_decrease' = 0, 'criterion' = 'gini', 'min_samples_split' = 11, 'n_estimators' = 47].

**Table 5.** The parameter space of RF.

| Parameters | 'n_estimators' | 'criterion' | 'max_depth' | 'max_features' | 'min_samples_split' |
|---|---|---|---|---|---|
| Selected range | [25, 50] | ['gini', 'entropy'] | [4, 26] | [1, 10] | [4, 20] |

The GBDT model identifies a total of eight parameter values (Table 6). As with the RF model, 'n_estimators' have the greatest impact on the modeling results. 'learning_rate' requires a trade-off with 'n_estimators'. $H(x_i)$ grows faster and requires fewer 'n_estimators' when 'learning_rate' increases; 'subsample' controls the proportion of samples that are put back into

the extraction, while other parameters play a similar role to RF. The final combination of parameters is ['learning_rate' = 0.16, 'loss' = 'exponential', 'max_depth' = 5, 'max_features' = 'sqrt', 'min_impurity_decrease' = 0.65, 'n_estimators' = 151, 'subsample' = 0.60].

**Table 6.** The parameter space of GBDT.

| Parameters | 'n_estimators' | 'learning_rate' | 'criterion' | 'loss' |
|---|---|---|---|---|
| Selected range | [130, 160] | [0.05, 2.05] | ['friedman_mse', 'squared_error'] | ['deviance', 'log_loss', 'exponential'] |
| Parameters | 'max_depth' | 'subsample' | 'max_features' | 'min_impurity_decrease' |
| Selected range | [5, 13] | [0.4, 0.8] | ['log2','sqrt',4,8,'auto'] | [0.4, 3.0] |

The experimental results show that the ensemble learning method obtains a higher prediction accuracy than the base classifier. SWA can better alleviate the overfitting and reduce bias. The four basic classifiers used in this study differ significantly in terms of classification effect and overfitting, and the greater the difference between the models, the better the results obtained by fusion will be. The final combination of weight values for SWA is [0.0014, 0.2866, 0.0404, 0.6716]. Models with better classification performance (GBDT-TPE) and fitting effectiveness (SVM-TPE) are given higher weight values. At the same time, models with weak classification performance (LR-TPE) and severe overfitting (RF-TPE) are given lower weight values. The combination of weights corresponds to the results of the analysis in Section 4.2. The above analysis shows that the performance improvement brought by the SWA is a guideline for decision-makers to analyze landslide susceptibility based on heuristic experimental results.

### 5.3. Exploration of Model Generalization Ability

To discuss the generalization ability of SWA, the four base models retain their original parameters in Section 5.2 and SWA is implemented on this basis. This study selects two scholars' publicly available landslide inventory datasets for testing. Firstly, the dataset of landslides in Yanshan County, China (Figure 11a), was made public by Fang et al. [71]. The basic geomorphology of Yanshan County is characterized by a high south and a low north. This dataset contains 380 historical landslide hazard sites, all of which are shallow landslides.

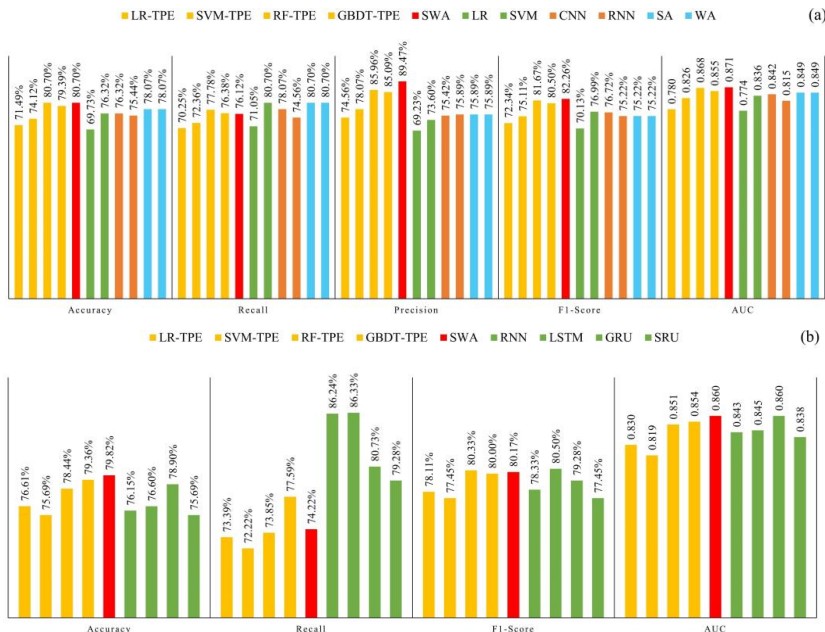

**Figure 11.** The model applicability of cross-regional test: (**a**) Yanshan country, China; (**b**) Yongxin country, China.

For the accuracy of a single model, SVM-TPE slightly decreases compared to SVM, but the other three models outperform LR, CNN, and RNN. In terms of Recall, the overall Recall value of the five selected models is lower than that of the model in the original paper. However, in terms of Precision, the Precision of five selected models is much higher than that of the original paper model. In the original paper, Simple averaging (SA) and Weighted averaging (WA) achieved the same performance, which indicates that the WA method failed to work. The SWA method in this paper achieves 80.70% Accuracy with AUC = 0.871, which is better than the classification performance of the original paper.

Wang et al. [72] selected four recurrent neural networks (RNNs) (Figure 11b) for LSM in Yongxin County, China. The geomorphology of Yongxin County is dominated by mountains and hills, with the terrain high in the north and south, low in the center, and tilting from the north and south to the center. The dataset contains 364 historical landslide hazard sites. A proportion of 70% of these landslides are rotational landslides and 30% of them belong to translational landslides. The accuracy of the selected model in this paper is higher than those of the four RNNs. But the Recall of all RNNs is higher than that of the selected model in this paper. The AUC of SWA is 0.860, which achieves the best classification of the original paper. In summary, BO is not only good for model effect improvement but can also improve the generalization ability of the model. SWA can overcome geographical differences across study regions and reconcile differences in classification performance over the base classifier, thereby improving the applicability of test results. Although SWA obtains good performance after cross-regional tests, this study only combines landslide predisposing factors with the algorithm to maximize the prediction performance [73,74], without mentioning the difference between landslide susceptibility and the real geographic environment. Landslide susceptibility zoning needs to consider the incomplete bias of landslide samples from roads and built-up areas in order to rationalize disaster prevention and control.

## 6. Conclusions

Based on 4 ML models (LR, SVM, RF, and GBDT) widely used in LSM, combined with BO, an ensemble-learning idea is introduced to establish 5 models of LR-TPE, SVM-TPE, RF-TPE, GBDT-TPE, and SWA for LSM in Yanbian Prefecture. The SWA method combines the advantages of individual classifiers and achieves better classification performance. The main conclusions based on the experimental results are as follows:

(i) All models achieve good LSM in the study area, with similar trends in the spatial distribution of susceptibility classes.

(ii) Secondly, SWA achieves an effective improvement in all metrics (Accuracy, Precision, Recall, F1-Score, Kappa coefficient, and AUC). In terms of the generalization ability of the model, SWA still performs well on small sample datasets and is highly applicable. The SWA method can maintain good stability while increasing the complexity of the model.

(iii) To visualize the contribution of the elements to the classification results in the modeling process, the SHAP method is selected for analysis in this paper, which results in the maximum influence of NDVI on the occurrence of landslides.

(iv) In summary, combining ensemble learning theory with ML models will make the models highly applicable and have the potential to produce reliable LSM. SWA is 19.39% and 19.94% in the very low and very high regions, respectively, and 20.45%, 20.19%, and 20.04% in the low, medium, and high regions, respectively.

In short, the results of this work can produce high-quality LSMs that can help managers make sound land spatial planning and reduce the hazards caused by landslides.

**Author Contributions:** Conceptualization, H.T. and C.W.; data curation, H.T., C.W. and S.A.; formal analysis, H.T.; investigation Q.W.; software, H.T.; writing—original draft preparation, H.T. and C.W.; writing—review and editing, H.T., S.A., and C.J. All authors have read and agreed to the published version of the manuscript.

**Funding:** This work was supported by the National Natural Science Foundation of China (grant number 41972267).

**Data Availability Statement:** Data sharing not applicable.

**Acknowledgments:** We would like to thank anonymous reviewers for their constructive and insightful comments and suggestions on the earlier version.

**Conflicts of Interest:** The authors declare no conflict of interest.

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
