# Peer review of "A Novel Heterogeneous Ensemble Framework Based on Machine Learning Models for Shallow Landslide Susceptibility Mapping"

_remotesensing, doi:10.3390/rs15174159_

Round 1

Reviewer 1 Report

General comments 
This study proposes an ensemble framework for Landslide susceptibility mapping (LSM) in Yanbian Prefecture. In this paper, 4 of the most representative Machine learning (ML) models, namely, logistic regression (LR), support vector machine (SVM), random forest (RF), and gradient boosting decision tree (GBDT), are selected as single classifiers. This study suggested combining ensemble learning theory with ML models to produce reliable LSM. Combined with Bayesian optimization (BO) to propose a stratified weighted averaging (SWA) framework can give better performance as compared to standard and commonly a single ML which suffer from limitations such as overfitting and unreliable accuracy. 

This article is based on multisource geographic information data and adopts an ensemble learning approach to effectively complete the evaluation of landslide susceptibility in Yanbian Prefecture, China. The proposed method overcomes the geographical differences in landslide susceptibility analysis of a single model and has a strong generalization ability, enabling accurate and reliable assessment results. The topic of this thesis is well within the field, and the proposed method effectively addresses the problem of overfitting and unreliable accuracy of a single model for landslide susceptibility mapping. In the end, Bayesian optimization was used to search for combinations of weight values, taking into account the accuracy while also improving efficiency. The research presented within the manuscript is reasonable and contributes to help managers make sound land spatial planning and how to reduce the hazards caused by landslides. 

Overall manuscript is well written. Statistical analysis performed by the authors is robust. 

Specific comments and suggestions:
1~How the lithology data was selected, it seems the author adopted raster dataset rather than the lithology distribution on the geology map

2~Line 193, why has Z-Score normalization been adopted for data processing, and what will be the impact on the results?

3~Citation 42, SVM has been widely used for landslide susceptibility analysis, please add more appropriate citations 

4~In section 3.3.2, the predicted probability of SWA is compared with the judgment threshold to categorize, whether the judgment threshold here is 0.5 or also tuned as a hyperparameter like the weight value, please specify

5~Logistic regression and random forest in the selected model can reflect the effect of selected elements on landslide occurrence. Please explain more details on SHAP method, and give more reason why this method was selected?

6~Please provide the specific formula for calculating Shapely values

7~Lines 470~478, please add the category to which the selected elements belong, e.g. topography, hydrology, ecology, etc., and please streamline the narrative in this section.

8~Two other regions were selected for model generalization capability exploration, please describe the topographic and geomorphic conditions of the two study regions to support the applicability of SWA

9~In section 3.3.2, it is important to point out that the value of the judgment threshold is taken.

10~Authors suggest that ensemble framework includes 4 of the most representative ML models, which can provide an effective landslide risk assessment. I would suggest author give more clues on whether there are other methods could potentially be used for such analysis.

Author Response

Dear reviewer,

   Firstly, we would like to express our heartfelt thanks to you for your hard work on our manuscript. All your comments have helped us a lot in revising our manuscript. Below is our point-to-point response to your comments. Also I attached the Word document. Please see the attachment

1-How the lithology data was selected, it seems the author adopted raster dataset rather than the lithology distribution on the geology map

Answer: First of all, the lithology classification on the geological map is very detailed, and the lithology classification of Yanbian can reach more than 20 kinds, and too many kinds of classifications can easily lead to the high dimensionality of a single feature during training, which has an adverse effect on the classification effect. On the other hand, the lithology raster data is classified according to the physical and chemical properties of lithology, which is simple and practical.

In addition, the resolution of the lithology raster is 250m, which is better than the data accuracy of the geologic map.

2-Line 193, why has Z-Score normalization been adopted for data processing, and what will be the impact on the results?

Answer: The training samples have been reclassified and coded in ArcGIS, all features are discrete, and the Z-score normalization makes the samples approximately obey the normal distribution, which makes the samples statistically significant and facilitates the training of the algorithm.

3-Citation 42, SVM has been widely used for landslide susceptibility analysis, please add more appropriate citations

Answer: The citation has been changed.

Kumar, D., Thakur, M., Dubey, C. S., & Shukla, D. P. (2017, 2017/10/15/). Landslide susceptibility mapping & prediction using Support Vector Machine for Mandakini River Basin, Garhwal Himalaya, India. Geomorphology, 295, 115-125. https://doi.org/10.1016/j.geomorph.2017.06.013

4-In section 3.3.2, the predicted probability of SWA is compared with the judgment threshold to categorize, whether the judgment threshold here is 0.5 or also tuned as a hyperparameter like the weight value, please specify

Answer: The judgment thresholds are searched as hyperparameters with the same weight values as well. Already added in Figure5 and Line356.

5-Logistic regression and random forest in the selected model can reflect the effect of selected elements on landslide occurrence. Please explain more details on SHAP method, and give more reason why this method was selected?

Answer: The SHAP method takes the weighted average of the marginal contribution of a feature over a subset of all feature combinations as the contribution of that feature, namely Shapley Value. When interpreting a machine learning model, the Shapley value can be interpreted as the importance of the input feature's contribution to the model's predicted value. The higher the Shapley value, the greater the contribution of the input feature to the predicted result. While logistic regression and random forests give the importance of the corresponding features, they do not directly reflect the degree to which the features affect the results.

6-Please provide the specific formula for calculating Shapely values

Answer: See Line 487-491 for the formula and explanation of the parameters.

7-Lines 470~478, please add the category to which the selected elements belong, e.g. topography, hydrology, ecology, etc., and please streamline the narrative in this section.

Answer: Mentioned parts have been rewritten, see Lines 491-505 for details.

8-Two other regions were selected for model generalization capability exploration, please describe the topographic and geomorphic conditions of the two study regions to support the applicability of SWA

Answer: The basic geomorphology of Yanshan County is characterized by a high south and a low north. The geomorphology of Yongxin County is dominated by mountains and hills, with the terrain high in the north and south, low in the center, and tilting from the north and south to the center. Added at Line 584 and Line 599.

9-In section 3.3.2, it is important to point out that the value of the judgment threshold is taken.

Answer: judgment threshold = 0.5688. Already added in Line358.

10-Authors suggest that ensemble framework includes 4 of the most representative ML models, which can provide an effective landslide risk assessment. I would suggest author give more clues on whether there are other methods could potentially be used for such analysis.

Answer: Supplements to this section can be found in Section 1, Lines 44-69.

Reviewer 2 Report

Thank you very much for inviting me to read this paper. I'm sorry for the late response, because I was on a field trip.

Landslide susceptibility mapping (LSM) plays a key role in landslide hazard management. This paper proposes an ensemble framework for landslide susceptibility prediction. The paper is readability and fit to this journal. But I have some comments before it is accepted.

1. Landslide inventories are the basis for landslide susceptibility studies. The authors stated that the landslide inventory was obtained from the Department of Natural Resources of Jilin Province. But detailed information on the landslide, such as the volume, the depth and the trigger factors of the landslide is missing.

2. The method and principle of obtaining the non-landslide points are missing.

3. According to Table 2, the proposed SWA method does not seem to have significantly better performance than the traditional method.

4. It is recommended that Figure 9 be divided into two graphs, one graph is ROC curves for the training database, and the other is ROC curves for the test database. 10 ROC curves in one figure are difficult to read.

5. The landslides in the training database are mainly shallow landslides (depth less than 10m). What is the landslide depth in the validation database (landslide datasets from Yanshan country and Yongxin country)?

Author Response

Dear reviewer,

   Firstly, we would like to express our heartfelt thanks to you for your hard work on our manuscript. All your comments have helped us a lot in revising our manuscript. Below is our point-to-point response to your comments. Also I attached the word document. Please see the attachment.

1-Landslide inventories are the basis for landslide susceptibility studies. The authors stated that the landslide inventory was obtained from the Department of Natural Resources of Jilin Province. But detailed information on the landslide, such as the volume, the depth and the trigger factors of the landslide is missing.

Answer: The number of landslides is too large to list, but the overall scale is summarized as shallow landslides with a thickness of 0.5‒10 m., triggered by rainfall, as already mentioned in the text. The historical hazard point years for the landslide inventory are 2010-2020 and have been confirmed by field research. This section is described in Line 113-120.

2-The method and principle of obtaining the non-landslide points are missing.

Answer: First of all, based on the spatial distribution of landslides, it is known that the extent of landslides is limited, and the vast majority of places did not experience landslides. In addition, an appropriate expansion of the number of negative samples is also beneficial to the classification results of the model, which has been explored by scholars in the article mentioned in the citation at 20. Yang, C., Liu, L.-L., Huang, F., Huang, L., & Wang, X.-M. (2022). Machine learning-based landslide susceptibility assessment with optimized ratio of landslide to non-landslide samples. Gondwana Research. https://doi.org/10.1016/j.gr.2022.05.012

Therefore, the scale selected in this paper is consistent with the actual geographic environment and also helps in model training.

The non-landslide samples were selected at a ratio of 1:1.15 (appropriately expanding the number of negative samples) while randomly selected outside the 400m buffer zone of the landslide point.

3-According to Table 2, the proposed SWA method does not seem to have significantly better performance than the traditional method.

Answer: The improvement in model performance consists of two aspects. On the one hand, in terms of evaluation metrics, such as accuracy, AUC value, and recall, SWA has seen a 3% improvement over the worst performing single model, as well as a 2% improvement in classification effectiveness. On the other hand, there is the fight against overfitting. Landslide susceptibility mapping requires reliable results, and SWA has the best fitting performance of all models. SWA does not appear to be a significant improvement (e.g., increasing accuracy by 5%), but SWA improvements are effective and easy to generalize.

4-It is recommended that Figure 9 be divided into two graphs, one graph is ROC curves for the training database, and the other is ROC curves for the test database. 10 ROC curves in one figure are difficult to read.

Answer: The ROC curves have been split into two graphs for the training and test sets for easy reading. See Figure 9 for details.

5- The landslides in the training database are mainly shallow landslides (depth less than 10m). What is the landslide depth in the validation database (landslide datasets from Yanshan country and Yongxin country)?

Answer: The Yanshan dataset contains 380 historical landslide hazard sites, all of which are shallow landslides. 70% of Yongxin landslides are rotational landslide and 30% of them belong to transla-tional landslide. This detail has been added to Section 5.3.

Reviewer 3 Report

General comments

In the paper “A novel heterogeneous ensemble framework based on machine learning models for shallow landslide susceptibility mapping”, the author uses an ensemble framework based on Bayesian optimization to perform regional-scale landslide susceptibility mapping. Many quantitative indicators are used to compare with numerous classical machine learning models. The line of reasoning is acceptable, while the methodology in this work is regularly used in statistical landslide susceptibility studies, as is the data analysis process. There are several comments below for the consideration of the authors.

Specific comments

1. The authors need to clarify what the contribution and added value of this article is. The Bayesian Optimal Ensemble framework used in the article has been applied in landslide susceptibility studies, such as the following. The authors need to state the advances and innovations compared to the existing studies.

Sameen, M. I., Pradhan, B., & Lee, S. (2020). Application of convolutional neural networks featuring Bayesian optimization for landslide susceptibility assessment. Catena, 186, 104249.

Zeng, T., Wu, L., Peduto, D., Glade, T., Hayakawa, Y. S., & Yin, K. (2023). Ensemble learning framework for landslide susceptibility mapping: Different basic classifier and ensemble strategy. Geoscience Frontiers, 101645.

Rong, G., Alu, S., Li, K., Su, Y., Zhang, J., Zhang, Y., & Li, T. (2020). Rainfall induced landslide susceptibility mapping based on Bayesian optimized random forest and gradient boosting decision tree models—A case study of Shuicheng County, China. Water, 12(11), 3066.

2. The landslide inventory is critical to this work, but the authors do not provide enough details. Were these landslides triggered by earthquake or rainfall or human activities? The timeframe of the landslide? All information is relevant to the selection of appropriate influencing factors and needs to be elaborated. There is no related representation in the current text at all. It is suggested that the author expand the relevant statements.

3. The selection of the landslide influencing factors requires further clarification. The authors also need to clearly explain, especially for the choice of NDVI. The remote sensing image used for calculating NDVI before or after the landslide occurred? Please consider that some variables might describe the conditions after landslide occurrence and the conditions before (the NDVI). If you want to identify susceptible terrain you should only use variables which do represent the conditions before slope failure, otherwise you are going towards “landslide detection” (i.e. the NDVI for example shows where a landslide happened in the past, but the location where future landslides will happen do not show yet such a specific post-landslide NDVI). The range of NDVI values should be between -1 and 1, why does the range go far beyond that in the authors' influencing factor map? The same problem exists with land cover, precipitation and SPI.

4. The number of non-landslide samples generated randomly by the authors is not in the commonly used 1:1 ratio to the number of landslide samples. For clarification, could the authors explain the reasons for the choice of the number of non-landslide points and the possible effects.

5. It is not appropriate to compare the two susceptibility maps based on the natural breaks method. The quantile method should be used to classify the two susceptibility maps and then they can be compared. The authors assessed the performance of the models based on the results of the one validation dataset alone. It is suggested that the authors may refer to the existing studies (Goetz et al., 2015) cited in the paper and further calculate each metric using methods such as spatial cross-validation and non-spatial cross-validation as a basis for checking the robustness of the results and conclusions obtained. In addition, the same prediction performance of the AUROC does not necessarily imply a consistent spatial pattern of landslide susceptibility. It is also necessary to take into account aspects such as the appearance of the generated landslide susceptibility distributions and their geomorphological plausibility. In this way, more reliable conclusions can be demonstrated.

Goetz, J. N., Brenning, A., Petschko, H., & Leopold, P. (2015). Evaluating machine learning and statistical prediction techniques for landslide susceptibility modeling. Computers & geosciences, 81, 1-11.

Steger, S., Brenning, A., Bell, R., Petschko, H., & Glade, T. (2016). Exploring discrepancies between quantitative validation results and the geomorphic plausibility of statistical landslide

6. The distribution of landslide inventory data mapped by the authors shows that there is incomplete bias. The landslide samples are mainly distributed along roads, and close to built-up areas. The authors need to take into account whether this incomplete bias spatial distribution affects the modelling? It is suggested that the authors should at least add a few sentences to the discussion section by referring to the following documents.

Lima, P., Steger, S., & Glade, T. (2021). Counteracting flawed landslide data in statistically based landslide susceptibility modelling for very large areas: a national-scale assessment for Austria. Landslides, 18(11), 3531-3546.

Lin, Q., Lima, P., Steger, S., Glade, T., Jiang, T., Zhang, J., ... & Wang, Y. (2021). National-scale data-driven rainfall induced landslide susceptibility mapping for China by accounting for incomplete landslide data. Geoscience Frontiers, 12(6), 101248.

Steger, S., Mair, V., Kofler, C., Pittore, M., Zebisch, M., & Schneiderbauer, S. (2021). Correlation does not imply geomorphic causation in data-driven landslide susceptibility modelling–Benefits of exploring landslide data collection effects. Science of the total environment, 776, 145935.

Author Response

Dear reviewer,

   Firstly, we would like to express our heartfelt thanks to you for your hard work on our manuscript. All your comments have helped us a lot in revising our manuscript. Below is our point-to-point response to your comments. Also I attached the word document. Please see the attachment

1-The authors need to clarify what the contribution and added value of this article is. The Bayesian Optimal Ensemble framework used in the article has been applied in landslide susceptibility studies, such as the following. The authors need to state the advances and innovations compared to the existing studies.

Answer: Unlike previous related studies, this study obtained a set of model optimal parameter combinations after BO of a single model, and model derivation was performed on top of this to search for the best combination of weight values. On the one hand, the number of discriminative models is increased to improve the accuracy of prediction, and on the other hand, the computational process is simple and expandable.

This portion of the narrative is supplemented in Lines 84-88 with a citation to a paper you recommend cited in Article.

  1. Zeng, T., Wu, L., Peduto, D., Glade, T., Hayakawa, Y. S., & Yin, K. (2023, 2023/11/01/). Ensemble learning framework for landslide susceptibility mapping: Different basic classifier and ensemble strategy. Geoscience Frontiers, 14(6), 101645. https://doi.org/10.1016/j.gsf.2023.101645

2-The landslide inventory is critical to this work, but the authors do not provide enough details. Were these landslides triggered by earthquake or rainfall or human activities? The timeframe of the landslide? All information is relevant to the selection of appropriate influencing factors and needs to be elaborated. There is no related representation in the current text at all. It is suggested that the author expand the relevant statements.

Answer: The trigger for landslides is rainfall, and the timeframe of the survey is the historical hazard sites between 2010 and 2020 as well as the locations of the sites obtained from the field surveys. This section is described in Line 115-120.

A detailed description of the influencing factors is given in Line 159-195.

3-The selection of the landslide influencing factors requires further clarification. The authors also need to clearly explain, especially for the choice of NDVI. The remote sensing image used for calculating NDVI before or after the landslide occurred? Please consider that some variables might describe the conditions after landslide occurrence and the conditions before (the NDVI). If you want to identify susceptible terrain you should only use variables which do represent the conditions before slope failure, otherwise you are going towards “landslide detection” (i.e. the NDVI for example shows where a landslide happened in the past, but the location where future landslides will happen do not show yet such a specific post-landslide NDVI). The range of NDVI values should be between -1 and 1, why does the range go far beyond that in the authors' influencing factor map? The same problem exists with land cover, precipitation and SPI.

Answer: The data source of NDVI was selected from the 30m sub-provincial NDVI dataset for 1986-2021 published by the team of researcher Xinliang Xu at the Institute of Geographic Sciences and Resources of the Chinese Academy of Sciences.

https://www.resdc.cn/,DOI:10.12078/2022030801

The selected NDVI values are for the year 2020, while researcher Xinliang Xu released the Google Earth Engine code for data acquisition, and the extracted NDVI was enlarged by 10,000 times in order to save storage units. In order to eliminate unnecessary misunderstandings, this has been corrected in Figure 2n

Rainfall values are magnified by a factor of 10, corrected to eliminate unnecessary misunderstandings in Figure 2p.

Land cover data are labeled with type in Figure 2n.

SPI calculations from reference 38, calculated without error in Figure 2f

  1. Sameen, M. I., Pradhan, B., & Lee, S. (2020). Application of convolutional neural networks featuring Bayesian optimization for landslide susceptibility assessment. Catena, 186, 13, Article 104249. https://doi.org/10.1016/j.catena.2019.104249

4-The number of non-landslide samples generated randomly by the authors is not in the commonly used 1:1 ratio to the number of landslide samples. For clarification, could the authors explain the reasons for the choice of the number of non-landslide points and the possible effects.

Answer: First of all, based on the spatial distribution of landslides, it is known that the extent of landslides is limited, and the vast majority of places did not experience landslides. In addition, an appropriate expansion of the number of negative samples is also beneficial to the classification results of the model, which has been explored by scholars in the article mentioned in the citation at 20. Yang, C., Liu, L.-L., Huang, F., Huang, L., & Wang, X.-M. (2022). Machine learning-based landslide susceptibility assessment with optimized ratio of landslide to non-landslide samples. Gondwana Research. https://doi.org/10.1016/j.gr.2022.05.012

Therefore, the scale selected in this paper is consistent with the actual geographic environment and also helps in model training.

The non-landslide samples were selected at a ratio of 1:1.15 (appropriately expanding the number of negative samples) while randomly selected outside the 400m buffer zone of the landslide point.

5-It is not appropriate to compare the two susceptibility maps based on the natural breaks method. The quantile method should be used to classify the two susceptibility maps and then they can be compared. The authors assessed the performance of the models based on the results of the one validation dataset alone. It is suggested that the authors may refer to the existing studies (Goetz et al., 2015) cited in the paper and further calculate each metric using methods such as spatial cross-validation and non-spatial cross-validation as a basis for checking the robustness of the results and conclusions obtained. In addition, the same prediction performance of the AUROC does not necessarily imply a consistent spatial pattern of landslide susceptibility. It is also necessary to take into account aspects such as the appearance of the generated landslide susceptibility distributions and their geomorphological plausibility. In this way, more reliable conclusions can be demonstrated.

Answer: The reclassification has been carried out according to the quartile method in accordance with your comments, and the changes in the classification results are shown in the Line 419-432, Figure 7 and Figure 8.

With regard to the metrics calculated in the text, they were obtained after 5 rounds of cross-validation, with references to the literature you recommended, see Line 448 for details.

Goetz, J. N., Brenning, A., Petschko, H., & Leopold, P. (2015, Aug). Evaluating machine learning and statistical prediction techniques for landslide susceptibility modeling [Article]. Computers & Geosciences, 81, 1-11. https://doi.org/10.1016/j.cageo.2015.04.007

The discussion of geomorphic plausibility is at Line 432-438

6-The distribution of landslide inventory data mapped by the authors shows that there is incomplete bias. The landslide samples are mainly distributed along roads, and close to built-up areas. The authors need to take into account whether this incomplete bias spatial distribution affects the modelling? It is suggested that the authors should at least add a few sentences to the discussion section by referring to the following documents.

Answer: There is spatial bias in the distribution of landslide susceptibility with respect to built-up areas, and based on the literature you recommended, I have placed the relevant discussion in Line 609-614.

  1. Steger, S., Mair, V., Kofler, C., Pittore, M., Zebisch, M., & Schneiderbauer, S. (2021, 2021/07/01/). Correlation does not imply geomorphic causation in data-driven landslide susceptibility modelling – Benefits of exploring landslide data collection effects. Science of The Total Environment, 776, 145935.https://doi.org/10.1016/j.scitotenv.2021.145935
  2. Lima, P., Steger, S., & Glade, T. (2021, 2021/11/01). Counteracting flawed landslide data in statistically based landslide susceptibility modelling for very large areas: a national-scale assessment for Austria. Landslides, 18(11), 3531-3546. https://doi.org/10.1007/s10346-021-01693-

Reviewer 4 Report

The manuscript A novel heterogeneous ensemble framework based on machine learning models for shallow landslide susceptibility mapping has an interesting technical novel and was well-structured.

Lines 33-34: Landslide susceptibility refers to the likelihood of landslides occurring in a target area based on local conditions and predisposing factors.

 Improve this discussion in the introduction, and cite papers. Examples:

DOI: 10.3390/rs11020196

DOI: 10.3390/rs12030502

DOI: 10.3390/IECG2022-13864

DOI: 10.1016/j.jrmge.2022.07.009

 —-------------------------

Lines 141-143: Normalized Difference Vegetation Index (NDVI) is calculated for each annual year of Landsat 5/8 remote sensing images based on Google Earth Engine (GEE), and then the maximum NDVI is obtained for the location of each image pixel.

What atmospheric correction method was used?

 In table 1 it says that the NDVI source is

Derived from Resource and Environment Science and Data Center (https://www.resdc.cn/) and in line 143, it mentions access to data by Google Earth Engine. What is the source of the data?

 —-------------------------

 Summarize the importance of landslide predisposing factors in conclusions. What are the most and least important in the analysis?

 —-------------------------

 Improve your understanding of using the suggested approach in relation to public policy, management, and monitoring (mentioned in the conclusions). I suggest creating a subtopic in a discussion about public policy implications.

 Some articles that may contribute to this discussion:

DOI: 10.1111/ele.12389

DOI: 10.1007/s10668-021-01293-4

DOI: 10.1016/j.ncon.2016.03.003

DOI: 10.1016/j.envres.2022.115155

DOI: 10.1016/j.jclepro.2019.119550

 —-------------------------

 I agree with the publication after minor reviews.

Author Response

Dear reviewer,

   Firstly, we would like to express our heartfelt thanks to you for your hard work on our manuscript. All your comments have helped us a lot in revising our manuscript. Below is our point-to-point response to your comments. Also I attached the word document. Please see the attachment.

1-Lines 33-34: Landslide susceptibility refers to the likelihood of landslides occurring in a target area based on local conditions and predisposing factors.

Answer: The discussion of landslide susceptibility has been modified here, as detailed in Line 33-38, with two citations from the literature you recommended.

  1. Ghorbanzadeh O, Blaschke T, Gholamnia K, Meena SR, Tiede D, Aryal J. Evaluation of Different Machine Learning Methods and Deep-Learning Convolutional Neural Networks for Landslide Detection. Remote Sensing. 2019; 11(2):196. https://doi.org/10.3390/rs11020196
  2. Chang, Z. L., Catani, F., Huang, F. M., Liu, G. Z., Meena, S. R., Huang, J. S., & Zhou, C. B. (2023, May). Landslide suscepti-bility prediction using slope unit-based machine learning models considering the heterogeneity of conditioning factors. Journal of Rock Mechanics and Geotechnical Engineering, 15(5), 1127-1143. https://doi.org/10.1016/j.jrmge.2022.07.009

2-Lines 141-143: Normalized Difference Vegetation Index (NDVI) is calculated for each annual year of Landsat 5/8 remote sensing images based on Google Earth Engine (GEE), and then the maximum NDVI is obtained for the location of each image pixel. What atmospheric correction method was used?

In table 1 it says that the NDVI source is derived from Resource and Environment Science and Data Center (https://www.resdc.cn/) and in line 143, it mentions access to data by Google Earth Engine. What is the source of the data?

Answer: FLAASH atmospheric correction method. This data set was obtained from the Google Earth Engine (GEE) code published by the data publisher at https://www.resdc.cn/,DOI:10.12078/2022030801

Regarding the data source of NDVI, this dataset is the 30m Chinese sub-provincial NDVI dataset for 1986-2021 published by Xinliang Xu, a research institute of the Institute of Geographic Sciences and Resources, Chinese Academy of Sciences. Since the NDVI values used for the study are for the year 2020, but the free data is limited to 2021. The dataset in this paper was obtained from the GEE code, so the original data source is labeled.

3-Summarize the importance of landslide predisposing factors in conclusions. What are the most and least important in the analysis?

Answer: See Lines 491-505 for a detailed discussion of the most important susceptibility factors.

4-Improve your understanding of using the suggested approach in relation to public policy, management, and monitoring (mentioned in the conclusions). I suggest creating a subtopic in a discussion about public policy implications. Some articles that may contribute to this discussion

Answer: Firstly, the topic of this thesis is based on the stratified weighted averaging framework for landslide susceptibility, so the main content of the discussion is about the analysis of the causes of landslides, the optimization of the model hyper-parameters, and the discussion of the model's generalization ability

Secondly, landslide susceptibility is the first step in the prevention and control of landslides, which includes the assessment of the hazard of landslides, the risk zoning of landslides. The determination of the risk zoning is only helpful for discussing the relevant prevention and control measures.

Finally, this paper is limited in length, and the parts that you proposed will be mentioned appropriately in the part of the discussion on the Section 5.1.

The specifics are in Line 504-510, along with 3 citations to the literature you recommend.

  1. Martinez, A. d. l. I., & Labib, S. M. (2023, 2023/03/01/). Demystifying normalized difference vegetation index (NDVI) for greenness exposure assessments and policy interventions in urban greening. Environmental Research, 220, 115155.https://doi.org/10.1016/j.envres.2022.115155
  2. Wong, C. P., Jiang, B., Kinzig, A. P., Lee, K. N., & Ouyang, Z. Y. (2015, Jan). Linking ecosystem characteristics to final ecosystem services for public policy [Review]. Ecology Letters, 18(1), 108-118. https://doi.org/10.1111/ele.12389
  3. He, Z., Xiao, L., Guo, Q., Liu, Y., Mao, Q., & Kareiva, P. (2020, 2020/04/01/). Evidence of causality between economic growth and vegetation dynamics and implications for sustainability policy in Chinese cities. Journal of Cleaner Production, 251, 119550.https://doi.org/10.1016/j.jclepro.2019.119550

Round 2

Reviewer 2 Report

  • The revised paper is acceptable.

Reviewer 3 Report

My concerns and suggestions were well-addressed.